# Invariant Learning via Probability of Sufficient and Necessary Causes

**Mengyue Yang[1], Zhen Fang[2], Yonggang Zhang[3]\*,**
**Yali Du[4], Furui Liu[5], Jean-Francois Ton[6], Jianhong Wang[7], Jun Wang[1]\***
[1]University College London, [2]University of Technology Sydney
[3]Hong Kong Baptist University, [4]King's College London
[5]Zhejiang Lab, [6]ByteDance Research, [7]University of Manchester
mengyue.yang.20@ucl.ac.uk, zhen.fang@uts.edu.au,
csygzhang@comp.hkbu.edu.hk, yali.du@kcl.ac.uk
liufurui@zhejianglab.com, jeanfrancois@bytedance.com
jianhong.wang@manchester.ac.uk,jun.wang@cs.ucl.ac.uk

## Abstract

Out-of-distribution (OOD) generalization is indispensable for learning models in the wild, where testing distribution typically unknown and different from the training. Recent methods derived from causality have shown great potential in achieving OOD generalization. However, existing methods mainly focus on the invariance property of causes, while largely overlooking the property of *sufficiency* and *necessity* conditions. Namely, a necessary but insufficient cause (feature) is invariant to distribution shift, yet it may not have required accuracy. By contrast, a sufficient yet unnecessary cause (feature) tends to fit specific data well but may have a risk of adapting to a new domain. To capture the information of sufficient and necessary causes, we employ a classical concept, the probability of sufficiency and necessary causes (PNS), which indicates the probability of whether one is the necessary and sufficient cause. To associate PNS with OOD generalization, we propose PNS risk and formulate an algorithm to learn representation with a high PNS value. We theoretically analyze and prove the generalizability of the PNS risk. Experiments on both synthetic and real-world benchmarks demonstrate the effectiveness of the proposed method. The detailed implementation can be found at the GitHub repository: https://github.com/ymy4323460/CaSN.

## 1 Introduction

The traditional supervised learning methods heavily depend on the in-distribution (ID) assumption, where the training data and test data are sampled from the same data distribution (Shen et al., 2021; Peters et al., 2016). However, the ID assumption may not be satisfied in some practical scenarios like distribution shift (Zhang et al., 2013; Sagawa et al., 2019), which leads to the failure of these traditional supervised learning methods. To relax the ID assumption, researchers have recently started to study a different learning setting called *out-of-distribution* (OOD) *generalization*. OOD generalization aims to train a model using the ID data such that the model generalizes well in the unseen test data that share the same semantics with ID data (Li et al., 2018b; Ahuja et al., 2021).

Recent works have proposed to solve the OOD generalization problem through the lens of causality (Peters et al., 2016; Pfister et al., 2019; Rothenhäusler et al., 2018; Heinze-Deml et al., 2018; Gamella

---

\*Corresponding Author: *csygzhang@comp.hkbu.edu.hk; jun.wang@cs.ucl.ac.uk*

37th Conference on Neural Information Processing Systems (NeurIPS 2023).

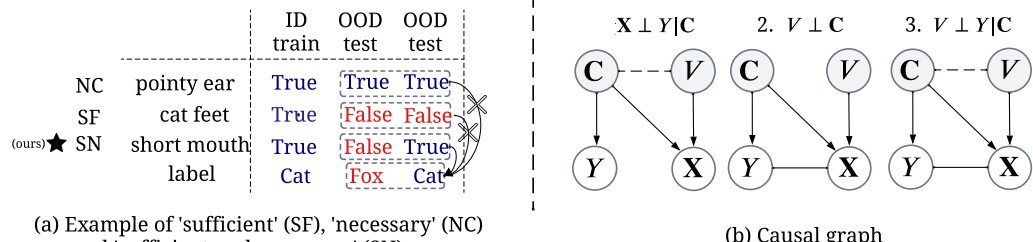

(a) Example of 'sufficient' (SF), 'necessary' (NC) and 'sufficient and necessary' (SN) causes

(b) Causal graph

Figure 1: (a) Examples for causal sufficiency and necessity in the cat classification. (b) The causal graph for OOD generalization problem. The arrows denote the causal generative direction and the dashed line connects the spurious correlated variables. Notations are formally defined in Section 2.1. & Heinze-Deml, 2020; Oberst et al., 2021; Chen et al., 2022a). These works focus on learning invariant representation, aiming to capture the cause of the labels. By learning this representation, one can bridge the gap between the ID training data and unknown OOD test data, and thus mitigate the negative impacts on the distribution shift between ID and OOD distributions. Among these works, invariant risk minimization (IRM) (Arjovsky et al., 2019) is the most representative method, targeting to identify invariant representation and classifier using a bi-level optimization algorithm. Following works, many efforts have been devoted to further extending the original invariant learning framework (Chen et al., 2022b; Ahuja et al., 2020a; Lu et al., 2021; Liu et al., 2021b; Lin et al., 2022).

Noticeably, the aforementioned invariant learning methods mainly focus on learning the invariant causal representation, which may contain non-essential information that is not necessary nor sufficient information (Pearl, 2009). In image classification tasks, necessity describes the label is not true if the features disappear, and sufficiency describes the presence of a feature helps us determine the correctness of the label. If the feature extractor only learns a representation that is invariant but fails to satisfy the sufficiency or necessity, the model's generalisation ability may deteriorate. As an illustrative example (see Figure 1(a)), suppose that the training data only contains images of cats with feet and that we are interested in learning a model for a cat prediction task. If the model captures the invariant information (feature) "cat feet", then the learned model is likely to make a mistake in the OOD data containing cats without "cat feet" features. The example "cat feet" demonstrates the representation contains sufficient but unnecessary causal information because using "cat feet" can predict the label "cat" but a cat image might not contain "cat feet". Analogously, there are also representations that are necessary but not sufficient (the feature "pointy ear" in Figure 1(a)). In Section 2.2, we present more examples to enhance the understanding of sufficiency and necessity.

This paper proposes achieving OOD generalization using *essential causal information*, which builds upon the probability of *necessity* and *sufficiency* (PNS) (Pearl, 2009). In this paper, we introduce the PNS risk. A low PNS risk implies that the representation contains both the necessary and sufficient causal information from the observation data with a high level of confidence. We provide some theoretical analysis that establishes the approximation of the risk on unseen test domains by the risk on source data. Based on these theoretical results, we discuss PNS risk in the context of a semantic separable representation space and propose an algorithm for learning the representation which contains the information of both sufficient and necessary causes from training data (ID data) under different causal assumptions in Figure 1(b). The main contributions of this paper are as follows:

Firstly, we propose a new learning risk—PNS risk—to estimate the sufficiency and necessity of information contained in the learned representation. Secondly, we theoretically analyze the PNS risk under OOD problem and bound the gap between PNS risk on the test domain distribution and the risk on source data. Lastly, we propose an algorithm that captures sufficient and necessary causal representation with low PNS risk on test domains. Experiments on synthetic and real-world benchmarks are conducted to show the effectiveness of the algorithm over state-of-the-art methods.

## 2 Preliminaries

### 2.1 Learning Setups

**Domains.** Let $\mathbf{X} \in \mathcal{X} \subset \mathbb{R}^D$ be the observable feature variable and $Y \in \mathcal{Y}$ be the label. In this paper, we mainly focus on binary classification task[2], i.e., the label space $\mathcal{Y} = \{0, 1\}$. $\mathcal{S}$ is a joint

---

[2]We extend the binary classification to multi-classification in experiments.

distribution $P_s(\mathbf{X}, Y)$ defined over $\mathcal{X} \times \mathcal{Y}$ in source domain. Equivalently, the unseen test domain is $\mathcal{T} := P_t(\mathbf{X}, Y)$. We also set $\mathcal{T}_{\mathbf{X}} := P_t(\mathbf{X})$ to be the marginal distribution over variable $\mathbf{X}$ on test domain. Similarly, $\mathcal{S}_{\mathbf{X}} := P_s(\mathbf{X})$ is the marginal distributions on source domain over $\mathbf{X}$.

**Assumption and model.** We conclude the causal graph in OOD generalization in Figure 1(b), inspired by the content (invariant) features and style (variant) features partition (Zhang et al., 2022). There are the invariant feature $\mathbf{C} \in \mathbb{R}^d$ and domain specific variable (i.e. domain indicator) $V \in \{1, \cdots, n\}$ A common *assumption* of OOD generalization is that there exists latent causal variable $\mathbf{C} \in \mathbb{R}^d$ that maintains the invariance property across domains (see Figure 1(b)), i.e., $P_s(Y|\mathbf{C} = \mathbf{c}) = P_t(Y|\mathbf{C} = \mathbf{c})$ (Arjovsky et al., 2019). Built upon this assumption, we define an invariant predictor by using a simple linear classifier $\mathbf{w} : \mathbb{R}^d \to \mathcal{Y}$ on the causal features to get label $y = \text{sign}(\mathbf{w}^\top \mathbf{c})$. Since the causal variable $\mathbf{C}$ cannot be directly observed, we infer $\mathbf{C}$ from the observational data $\mathbf{x} \sim \mathbf{X}$. Then, the invariant predictor with the invariant representation inference model is defined as below.

$$y = \text{sign}[\mathbb{E}_{\mathbf{c} \sim P_t(\mathbf{C}|\mathbf{X}=\mathbf{x})} \mathbf{w}^\top \mathbf{c}]. \tag{1}$$

## 2.2 Probability of Sufficient and Necessary Cause

Existing invariant learning strategies (Rojas-Carulla et al., 2018; Pfister et al., 2019; Arjovsky et al., 2019) only consider the invariant property. However, the invariant representation can be further divided into three parts, each containing the different sufficient and necessary causal information.

**(i) Sufficient but unnecessary causes $A$:** The cause $A$ leads to the effect $B$, but when observing the effect $B$, it is hard to confirm $A$ is the actual cause (See example in Figure 1(a)). **(ii) Necessary but insufficient causes $A$:** Knowing effect $B$ we confirm the cause is $A$, but cause $A$ might not lead to effect $B$. "pointy ear" in cat prediction is selected as a typical example. Because when the ear shape is not pointy, we can confirm it is not a cat. However, a fox has a similar ear shape to a cat. Thus "pointy ear" is not a stable feature to predict cats. **(iii) Necessary and sufficient causes $A$:** Knowing the effect $B$ confirms the cause $A$, while observing $A$ leads to $B$. In the cat and fox classification task, "short mouth" could be a necessary and sufficient cause. It is because the feature "short mouth" allows us to distinguish a cat from a fox, and when know there is a cat, "short mouth" must exist.

In order to learn invariant representations $\mathbf{C}$ contains both sufficient and necessary causal information, we refer to the concept of *Probability of Sufficient and Necessary* (PNS) (Chapter 9 in (Pearl, 2009)), which is formally defined as below.

**Definition 2.1** (Probability of Necessary and Sufficient (PNS) (Pearl, 2009))**.** Let the specific implementations of causal variable $\mathbf{C}$ as $\mathbf{c}$ and $\bar{\mathbf{c}}$, where $\bar{\mathbf{c}} \neq \mathbf{c}$. The probability that $\mathbf{C}$ is the necessary and sufficiency cause of $Y$ on test domain $\mathcal{T}$ is

$$\text{PNS}(\mathbf{c}, \bar{\mathbf{c}}) := \underbrace{P_t(Y_{do(\mathbf{C}=\mathbf{c})} = y \mid \mathbf{C} = \bar{\mathbf{c}}, Y \neq y)}_{\text{sufficiency}} P_t(\mathbf{C} = \bar{\mathbf{c}}, Y \neq y)$$

$$+ \underbrace{P_t(Y_{do(\mathbf{C}=\bar{\mathbf{c}})} \neq y \mid \mathbf{C} = \mathbf{c}, Y = y)}_{\text{necessity}} P_t(\mathbf{C} = \mathbf{c}, Y = y). \tag{2}$$

In the above definition, the notion $P(Y_{do(\mathbf{C}=\bar{\mathbf{c}})} \neq y|\mathbf{C} = \mathbf{c}, Y = y)$ means that we study the probability of $Y \neq y$ when we force the manipulable variable $\mathbf{C}$ to be a fixed value $do(\mathbf{C} = \bar{\mathbf{c}})$ (do-operator) given a certain factual observation $Y = y$ and $\mathbf{C} = \mathbf{c}$. The first and second terms in PNS correspond to the probabilities of sufficiency and necessity, respectively. Variable $\mathbf{C}$ has a high probability to be the sufficient and necessary cause of $Y$ when the PNS value is large. Computing the counterfactual probability is a challenging problem since collecting the counterfactual data is difficult, or even impossible in real-world systems. Fortunately, PNS defined on counterfactual distribution can be directly estimated by the data under proper conditions, i.e., Exogeneity and Monotonicity.

**Definition 2.2** (Exogeneity (Pearl, 2009))**.** Variable $\mathbf{C}$ is exogenous relative to variable $Y$ w.r.t. source and test domains $\mathcal{S}$ and $\mathcal{T}$, if the intervention probability is identified by conditional probability $P_s(Y_{do(\mathbf{C}=\mathbf{c})} = y) = P_s(Y = y|\mathbf{C} = \mathbf{c})$ and $P_t(Y_{do(\mathbf{C}=\mathbf{c})} = y) = P_t(Y = y|\mathbf{C} = \mathbf{c})$.

**Definition 2.3** (Monotonicity [3](Pearl, 2009))**.** $Y$ is monotonic relative to $X$ if and only if either $P(Y_{do(\mathbf{C}=\mathbf{c})} = y, Y_{do(\mathbf{C}=\bar{\mathbf{c}})} \neq y) = 0$ or $P(Y_{do(\mathbf{C}=\mathbf{c})} \neq y, Y_{do(\mathbf{C}=\bar{\mathbf{c}})} = y) = 0$

---

[3]We rewrite logic expression $Y_{do(\mathbf{C}=\bar{\mathbf{c}})} = \bar{y} \wedge Y_{do(\mathbf{C}=\mathbf{c})} = y$ is false or $Y_{do(\mathbf{C}=\bar{\mathbf{c}})} = y \wedge Y_{do(\mathbf{C}=\mathbf{c})} = \bar{y}$ is false in original Monotonicity by the probabilistic formulation.

The definition of Exogeineity describes the gap between the intervention and conditional distributions vanishes when $\mathbf{C}$ is exogenous relative to $Y$ and the definition of Monotonicity demonstrates the monotonic effective on $Y$ of causal variable $\mathbf{C}$. Based on Definitions 2.2 and 2.3, the identifiability of PNS in Definition 2.1 is described as the following lemma.

**Lemma 2.4** (Pearl (2009)). *If $\mathbf{C}$ is exogenous relative to $Y$, and $Y$ is monotonic relative to $\mathbf{C}$, then*

$$PNS(\mathbf{c}, \bar{\mathbf{c}}) = \underbrace{P_t(Y = y | \mathbf{C} = \mathbf{c})}_{sufficiency} - \underbrace{P_t(Y = y | \mathbf{C} = \bar{\mathbf{c}})}_{necessity}. \tag{3}$$

According to Lemma 2.4, the computation of PNS is feasible through the observation data under Exogeneity and Monotonicity. This allows us to quantify PNS when counterfactual data is unavailable. The proof of Lemma 2.4 is provided by Pearl (2009). Wang & Jordan (2021) further extend the proof by incorporating probabilistic computation, as opposed to the logical calculation used in Pearl (2009).

# 3 PNS Risk Modeling

This section presents the PNS-based risk for invariant learning in OOD problem. The risk on test domains is a PNS-value evaluator, which is bounded by the tractable risk on the training domain.

## 3.1 PNS Risk

In this section, we introduce the PNS risk, which is a PNS-value estimator. The risk estimates the PNS value of the representation distribution $P_t(\mathbf{C}|\mathbf{X} = \mathbf{x})$ inferred from $\mathbf{X}$ on an unseen test domain $\mathcal{T}$. The risk increases when the representation contains less necessary and sufficient information, which can be caused by data distribution shifts. The PNS risk is based on the definition of PNS$(\mathbf{c}, \bar{\mathbf{c}})$. As $\bar{\mathbf{c}}$ represents the intervention value, it is not necessary for it to be a sample from the same distribution as the causal variable $\mathbf{C}$. Thus, we define an auxiliary variable $\bar{\mathbf{C}} \in \mathbb{R}^d$ (same as the range of $\mathbf{C}$) and sample $\bar{\mathbf{c}}$ from its distribution $P_t(\bar{\mathbf{C}}|\mathbf{X} = \mathbf{x})$. In the learning method, we use the notations $P_t^\phi(\mathbf{C}|\mathbf{X} = \mathbf{x})$ and $P_t^\xi(\bar{\mathbf{C}}|\mathbf{X} = \mathbf{x})$ to present the estimated distributions, which are parameterized by $\phi$ and $\xi$, separately. Let $\mathrm{I}(A)$ be an indicator function, where $\mathrm{I}(A) = 1$ if $A$ is true; otherwise, $\mathrm{I}(A) = 0$. PNS risk based on Definition 2.1 and Lamma 2.4 is formally defined as Eq. (4) below.

$$\begin{aligned} R_t(\mathbf{w}, \phi, \xi) := \mathbb{E}_{(\mathbf{x}, y) \sim \mathcal{T}} \big[ &\mathbb{E}_{\mathbf{c} \sim P_t(\mathbf{C}|\mathbf{X}=\mathbf{x})} \mathrm{I}[\mathrm{sign}(\mathbf{w}^\top \mathbf{c}) \neq y] \\ &+ \mathbb{E}_{\bar{\mathbf{c}} \sim P_t(\bar{\mathbf{C}}|\mathbf{X}=\mathbf{x})} \mathrm{I}[\mathrm{sign}(\mathbf{w}^\top \bar{\mathbf{c}}) = y] \big]. \end{aligned} \tag{4}$$

As the identifiability result in Lemma 2.4 is based on the Exogeneity 2.2 and Monotonicity 2.3, we modify the original risk equation, Eq. (4), to ensure compliance with these conditions. Below, we provide Monotonicity measurement and discuss the satisfaction of Exogeneity in Section 4.3.

**Satisfaction of monotonicity.** We naturally introduce the measurement of Monotonicity into PNS risk by deriving an upper bound of Eq. (4), which is given below.

**Proposition 3.1.** *Given a test domain $\mathcal{T}$, we define the sufficient and necessary risks as:*

$$SF_t(\mathbf{w}, \phi) := \underbrace{\mathbb{E}_{(\mathbf{x}, y) \sim \mathcal{T}} \mathbb{E}_{\mathbf{c} \sim P_t^\phi(\mathbf{C}|\mathbf{X}=\mathbf{x})} \mathrm{I}[\mathrm{sign}(\mathbf{w}^\top \mathbf{c}) \neq y]}_{sufficiency\ term},$$

$$NC_t(\mathbf{w}, \xi) := \underbrace{\mathbb{E}_{(\mathbf{x}, y) \sim \mathcal{T}} \mathbb{E}_{\bar{\mathbf{c}} \sim P_t^\xi(\bar{\mathbf{C}}|\mathbf{X}=\mathbf{x})} \mathrm{I}[\mathrm{sign}(\mathbf{w}^\top \bar{\mathbf{c}}) = y]}_{necessity\ term},$$

*and let the Monotonicity measurement be*

$$M_t^{\mathbf{w}}(\phi, \xi) := \mathbb{E}_{(\mathbf{x}, y) \sim \mathcal{T}} \mathbb{E}_{\mathbf{c} \sim P_t^\phi(\mathbf{C}|\mathbf{X}=\mathbf{x})} \mathbb{E}_{\bar{\mathbf{c}} \sim P_t^\xi(\bar{\mathbf{C}}|\mathbf{X}=\mathbf{x})} \mathrm{I}[\mathrm{sign}(\mathbf{w}^\top \mathbf{c}) = \mathrm{sign}(\mathbf{w}^\top \bar{\mathbf{c}})],$$

*then we have*

$$R_t(\mathbf{w}, \phi, \xi) = M_t^{\mathbf{w}}(\phi, \xi) + 2SF_t(\mathbf{w}, \phi)NC_t(\mathbf{w}, \xi) \leq M_t^{\mathbf{w}}(\phi, \xi) + 2SF_t(\mathbf{w}, \phi). \tag{5}$$

The upper bound for PNS risk in Eq. (5) consists of two terms: (i) the evaluator of sufficiency $SF_t(\mathbf{w}, \phi)$ and (ii) the Monotonicity measurement $M_t^{\mathbf{w}}(\phi, \xi)$. In the upper bound, the necessary term $NC_t(\mathbf{w}, \xi)$ is considered to be absorbed into measurement of Monotonicity $M_t^{\mathbf{w}}(\phi, \xi)$. The minimization process of Eq. (4) on its upper bound (5) considers the satisfaction of Monotonicity.

## 3.2 OOD Generalization with PNS risk

In OOD generalization tasks, only source data collected from $\mathcal{S}$ is provided, while the test domain $\mathcal{T}$ is unavailable during the optimization process. As a result, it is not possible to directly evaluate the risk on the test domain, i.e. $R_t(\mathbf{w}, \phi, \xi)$. To estimate $R_t(\mathbf{w}, \phi, \xi)$, we have a two-step process: (i) Firstly, since the test-domain distribution $\mathcal{T}$ is not available during the training process, We aim to establish a connection between the risk on the test domain $R_t(\mathbf{w}, \phi, \xi)$ and the risk on the source domain $R_s(\mathbf{w}, \phi, \xi)$ in Theorem 3.2. (ii) Furthermore, in practical scenarios where only a finite number of samples are available, we demonstrate the bound of the gap between the expected risk on the domain distribution and the empirical risk on the source domain data in Theorem 3.3.

**Connecting the PNS risks, i.e., $R_t(\mathbf{w}, \phi, \xi)$ and $R_s(\mathbf{w}, \phi, \xi)$.** We introduce divergence measurement $\beta$ divergence (Ganin et al., 2016) and weigh the $R_s(\mathbf{w}, \phi, \xi)$ term by variational approximation. $\beta$ divergence measures the distance between domain $\mathcal{T}$ and $\mathcal{S}$, which is formally defined below.

$$\beta_k(\mathcal{T}\|\mathcal{S}) = \left[ \mathbb{E}_{(\mathbf{x},y)\sim\mathcal{S}} \left( \frac{\mathcal{T}(\mathbf{x},y)}{\mathcal{S}(\mathbf{x},y)} \right)^k \right]^{\frac{1}{k}}. \tag{6}$$

Based on $\beta_k(\mathcal{T}\|\mathcal{S})$, we connect the risks on the source and test domains by Theorem 3.2.

**Theorem 3.2.** *The risk on the test domain is bounded by the risk on the source domain, i.e.,*

$$R_t(\mathbf{w}, \phi, \xi) \leq \lim_{k\to+\infty} \beta_k(\mathcal{T}\|\mathcal{S})([M_s^{\mathbf{w}}(\phi,\xi)]^{1-\frac{1}{k}} + 2[SF_s(\mathbf{w},\phi)]^{1-\frac{1}{k}}) + \eta_{t\backslash s}(\mathbf{X},Y),$$

*where*

$$\eta_{t\backslash s}(\mathbf{X},Y) := P_t(\mathbf{X}\times Y \notin \mathrm{supp}(\mathcal{S})) \cdot \sup R_{t\backslash s}(\mathbf{w}, \phi, \xi).$$

*Here $\mathrm{supp}(\mathcal{S})$ is the support set of source domain distribution $P_s(\mathbf{X})$,*

$$R_{t\backslash s}(\mathbf{w}, \phi, \xi) := \mathbb{E}_{(\mathbf{x},y)\sim P_t(\mathbf{X}\times Y\notin\mathrm{supp}(\mathcal{S}))}\big[\mathbb{E}_{\mathbf{c}\sim P_t(\mathbf{C}|\mathbf{X}=\mathbf{x})}\mathrm{I}[\mathrm{sign}(\mathbf{w}^\top\mathbf{c}) \neq y]$$
$$+\mathbb{E}_{\bar{\mathbf{c}}\sim P_t(\bar{\mathbf{C}}|\mathbf{X}=\mathbf{x})}\mathrm{I}[\mathrm{sign}(\mathbf{w}^\top\bar{\mathbf{c}}) = y]\big].$$

In Theorem 3.2, $\eta_{t\backslash s}(\mathbf{X},Y)$ describes the expectation of worst risk for unknown area i.e. the data sample $(\mathbf{x},y)$ does not include in the source domain support set $\mathrm{supp}(\mathcal{S})$. Theorem 3.2 connects the source-domain risk and the test-domain risk. In the ideal case, where $\mathbf{C}$ is the invariant representation, i.e. $P_s(Y|\mathbf{C}=\mathbf{c}) = P_t(Y|\mathbf{C}=\mathbf{c})$, the bound is reformed as below.

$$R_t(\mathbf{w}, \phi, \xi) \leq \lim_{k\to+\infty} \beta_k(\mathcal{T}_{\mathbf{X}}\|\mathcal{S}_{\mathbf{X}})([M_s^{\mathbf{w}}(\phi,T)]^{1-\frac{1}{k}} + 2[SF_s(\mathbf{w},\phi)]^{1-\frac{1}{k}}) + \eta_{t\backslash s}(\mathbf{X},Y). \tag{7}$$

When the observations $\mathbf{X}$ in $\mathcal{S}$ and $\mathcal{T}$ share the same support set, the term $\eta_{t\backslash s}(\mathbf{X},Y)$ approaches to 0. In domain generalization tasks, the term $\beta_k(\mathcal{T}_{\mathbf{X}}|\mathcal{S}_{\mathbf{X}})$ is treated as a hyperparameter, as the test domain $\mathcal{T}_{\mathbf{X}}$ is not available during training. However, in domain adaptation tasks where $\mathcal{T}_{\mathbf{X}}$ is provided, $\beta_k(\mathcal{T}_{\mathbf{X}}|\mathcal{S}_{\mathbf{X}})$ and the test-domain Monotonicity measurement $M_t^{\mathbf{w}}(\phi,\xi)$ can be directly estimated. Further details of the discussion on domain adaptation are provided in Appendix A.3.

**Connecting empirical risk to the expected risk.** In most real-world scenarios where distribution $\mathcal{S}$ is not directly provided, we consider the relationship of expected risk on source domain distribution and empirical risk on source domain data $\mathcal{S}^n := \{(\mathbf{x}_i, y_i)\}_{i=1}^n$. We also define the empirical risks w.r.t. $\widehat{SF}_s(\mathbf{w},\phi), \widehat{M}_s^{\mathbf{w}}(\phi,\xi)$ as follows:

$$\widehat{SF}_s(\mathbf{w},\phi) := \mathbb{E}_{\mathcal{S}^n}\mathbb{E}_{\mathbf{c}\sim\hat{P}_s^\phi(\mathbf{C}|\mathbf{X}=\mathbf{x})}\mathrm{I}[\mathrm{sign}(\mathbf{w}^\top\mathbf{c}) \neq y],$$

$$\widehat{M}_s^{\mathbf{w}}(\phi,\xi) := \mathbb{E}_{\mathcal{S}^n}\mathbb{E}_{\mathbf{c}\sim\hat{P}_s^\phi(\mathbf{C}|\mathbf{X}=\mathbf{x})}\mathbb{E}_{\bar{\mathbf{c}}\sim\hat{P}_s^\xi(\bar{\mathbf{C}}|\mathbf{X}=\mathbf{x})}\mathrm{I}[\mathrm{sign}(\mathbf{w}^\top\mathbf{c}) = \mathrm{sign}(\mathbf{w}^\top\bar{\mathbf{c}})],$$

where $\hat{P}_s^\phi(\mathbf{C}|\mathbf{X}=\mathbf{x})$ and $\hat{P}_s^\xi(\bar{\mathbf{C}}|\mathbf{X}=\mathbf{x})$ describe the estimated distribution on dataset $\mathcal{S}^n$.

Then, we use PAC-learning (Shalev-Shwartz & Ben-David, 2014) tools to formulate the upper bound of gap between empirical risk and expected risk as a theorem below.

**Theorem 3.3.** *Given parameters $\phi$, $\xi$, for any $\mathbf{w}: \mathbb{R}^d \to \mathcal{Y}$, prior distribution $\pi_{\mathbf{C}} := P_s(\mathbf{C})$ and $\pi_{\bar{\mathbf{C}}} := P_s(\bar{\mathbf{C}})$ which make $\mathbb{E}_{\mathcal{S}}\mathrm{KL}(P_s^\phi(\mathbf{C}|\mathbf{X}=\mathbf{x})\|\pi_{\mathbf{C}})$ and $\mathbb{E}_{\mathcal{S}}\mathrm{KL}(P_s^\xi(\bar{\mathbf{C}}|\mathbf{X}=\mathbf{x})\|\pi_{\bar{\mathbf{C}}})$ both lower than a positive constant C, then with a probability at least $1 - \epsilon$ over source domain data $\mathcal{S}_n$,*

*(1)* $|SF_s(\mathbf{w}, \phi) - \widehat{SF}_s(\mathbf{w}, \phi)|$ *is upper bounded by*

$$\mathbb{E}_{\mathcal{S}^n} \mathrm{KL}(\hat{P}_s^\phi(\mathbf{C}|\mathbf{X} = \mathbf{x}) \| \pi_\mathbf{C}) + \frac{\ln(n/\epsilon)}{4(n-1)} + C.$$

*(2)* $|M_s^\mathbf{w}(\phi, \xi) - \widehat{M}_s^\mathbf{w}(\phi, \xi)|$ *is upper bounded by*

$$\mathbb{E}_{\mathcal{S}^n} \mathrm{KL}(\hat{P}_s^\phi(\mathbf{C}|\mathbf{X} = \mathbf{x}) \| \pi_\mathbf{C}) + \mathbb{E}_{\mathcal{S}^n} \mathrm{KL}(\hat{P}_s^\xi(\bar{\mathbf{C}}|\mathbf{X} = \mathbf{x}) \| \pi_{\bar{\mathbf{C}}}) + \frac{\ln(n/\epsilon)}{4(n-1)} + 2C.$$

Theorem 3.3 demonstrates that as the sample size increases and the terms with KL divergence decrease, the empirical risk on the source domain dataset becomes closer to the expected risk. Combining Theorems 3.2 and 3.3, we can evaluate the expected PNS risk on the test distribution using the empirical risk on the source dataset. In the next section, we present a representation learning objective based on the results of Theorems 3.2 and 3.3 and introduce the satisfaction of Exogeneity.

## 4   Learning to Minimizing PNS Risk

In this section, we propose a learning objective built upon the PNS risk that is used to capture the essential representation having a high PNS value from observational data.

### 4.1   The Semantic Separability of PNS

In Section 3, we present PNS risk and Monotonicity measurement. Furthermore, to ensure that finding interpretable representations is feasible, we need to make certain assumptions that the representation of the data retains its semantic meaning under minor perturbations. Specifically, we define the variable $\mathbf{C}$ as Semantic Separability relative to $Y$ if and only if the following assumption is satisfied:

**Assumption 4.1** ($\delta$-Semantic Separability)**.** For any domain index $d \in \{s, t\}$, the variable $\mathbf{C}$ is $\delta$-semantic separable, if for any $\mathbf{c} \sim P_d(\mathbf{C}|Y = y)$ and $\bar{\mathbf{c}} \sim P_d(\mathbf{C}|Y \neq y)$, the following inequality holds almost surely: $\|\bar{\mathbf{c}} - \mathbf{c}\|_2 > \delta$.

$\delta$-Semantic Separability refers to the semantic meaning being distinguishable between $\mathbf{c}$ and $\bar{\mathbf{c}}$ when the distance between them is large enough, i.e., $\|\bar{\mathbf{c}} - \mathbf{c}\|_2 > \delta$. This assumption is widely accepted because, without it, nearly identical values would correspond to entirely different semantic information, leading to inherently unstable and chaotic data. If $\mathbf{C}$ satisfies Assumption 4.1, then considering the PNS value in a small intervention, such as $\|\mathbf{c} - \bar{\mathbf{c}}\|_2 < \delta$, will lead failure in representation learning. Therefore, during the learning process, we add the penalty of $\|\mathbf{c} - \bar{\mathbf{c}}\|_2 > \delta$.

### 4.2   Overall Objective

Depending on the diverse selections of $P^\xi(\bar{\mathbf{C}}|\mathbf{X} = \mathbf{x})$, there are multiple potential PNS risks. In the learning process, we consider minimizing the risk in the worst-case scenario lead by $\bar{\mathbf{C}}$, i.e., the maximal PNS risk lead by the selection of $P^\xi(\bar{\mathbf{C}}|\mathbf{X} = \mathbf{x})$. Minimizing the upper bounds in Theorems 3.2 and 3.3 can be simulated by the following optimization process

$$\min_{\phi, \mathbf{w}} \max_{\xi} \quad \widehat{M}_s^\mathbf{w}(\phi, \xi) + \widehat{SF}_s(\mathbf{w}, \phi) + \lambda L_{\mathrm{KL}}, \quad \text{subject to} \quad \|\mathbf{c} - \bar{\mathbf{c}}\|_2 > \delta, \tag{8}$$

where $L_{\mathrm{KL}} := \mathbb{E}_{\mathcal{S}^n} \mathrm{KL}(\hat{P}_s^\phi(\mathbf{C}|\mathbf{X} = \mathbf{x}) \| \pi_\mathbf{C})) + \mathbb{E}_{\mathcal{S}^n} \mathrm{KL}(\hat{P}_s^\xi(\bar{\mathbf{C}}|\mathbf{X} = \mathbf{x}) \| \pi_{\bar{\mathbf{C}}}))$. The constraint $\|\mathbf{c} - \bar{\mathbf{c}}\|_2 > \delta$ is set because of the Semantic Separability assumption. We name the algorithm of optimizing Eq. (8) as CaSN (Causal Representation of Sufficiency and Necessity).

### 4.3   Satisfaction of Exogeneity

In the previous sections, we introduced an objective to satisfy monotonicity. Identifying PNS values not only needs to satisfy monotonicity but also exogeneity. In this part, we discuss the satisfaction of Exogeneity and provide the solution to find the representation under three causal assumptions below.

**Assumption 4.2.** The Exogeneity of $\mathbf{C}$ holds, if and only if the following invariant conditions are satisfied separately under three causal assumptions in Figure 1(b): (1) $\mathbf{X} \perp Y|\mathbf{C}$ (2) $\mathbf{C} \perp V$ (3) $V \perp Y|\mathbf{C}$ (for assumption in Figure 1(b).1, 2 and 3, respectively).

The above three assumptions are commonly accepted by the OOD generalization (Lu et al., 2021; Liu et al., 2021a; Ahuja et al., 2021). To satisfy the Exogeneity, we use different objective functions to identify $\mathbf{C}$ over three invariant causal assumptions. For Assumption 4.2 (1), we provide the following theorem showing the equivalence between optimizing Eq.8 and identifying invariant representation.

**Theorem 4.3.** *The optimal solution of learned $\mathbf{C}$ is obtained by optimizing the following objective (the key part of the objective in Eq. (8))*

$$\min_{\phi, \mathbf{w}} \widehat{SF}_s(\mathbf{w}, \phi) + \lambda \mathbb{E}_{\mathcal{S}^n} \mathrm{KL}(\hat{P}_s^\phi(\mathbf{C}|\mathbf{X} = \mathbf{x}) \| \pi_{\mathbf{C}})$$

*satisfies the conditional independence $\mathbf{X} \perp Y | \mathbf{C}$.*

Theorem 4.3, details of the proof are shown in Appendix E, indicates optimizing overall objective Eq. (8) implicitly makes $\mathbf{C}$ satisfy the property of Exogeneity under causal assumption $\mathbf{X} \perp Y | \mathbf{C}$. For Assumption 4.2 (2), to identify the invariant assumption $\mathbf{C} \perp V$ (Li et al., 2018b), we introduce the following Maximum Mean Discrepancy (MMD) penalty to the minimization process in Eq. (8),

$$L_{\mathrm{mmd}} = \sum_{v_i} \sum_{v_j} \mathbb{E}_{\mathbf{x_i} \sim P(\mathbf{X}|V=v_i)} \mathbb{E}_{\mathbf{c_i} \sim \hat{P}_s^\phi(\mathbf{C}|\mathbf{X}=\mathbf{x_i})} \mathbb{E}_{\mathbf{x_j} \sim P(\mathbf{X}|V=v_j)} \mathbb{E}_{\mathbf{c_j} \sim \hat{P}_s^\phi(\mathbf{C}|\mathbf{X}=\mathbf{x_j})} \|\mathbf{c_i} - \mathbf{c_j}\|_2 .$$

For Assumption 4.2 (3), to specify the representation of $\mathbf{C}$ and allows the Exogeneity when the assumption $V \perp Y | \mathbf{C}$ holds, we introduce the IRM-based (Arjovsky et al., 2019) penalty into Eq.(8).

$$L_{\mathrm{irm}} = \sum_v E_{(\mathbf{x}, y) \sim P_s(\mathbf{X}, Y|V=v)} \left\| \nabla_{w|w=1.0} \mathbb{E}_{\mathbf{c} \sim \hat{P}_s^\phi(\mathbf{C}|\mathbf{X}=\mathbf{x})} \mathrm{I}[\mathrm{sign}(\mathbf{w}^\top \mathbf{c}) \neq y] \right\|^2$$

Noticebly, to address the issue of invariant learning and satisfy the Exogeneity under Assumptions 4.2 (2) and (3), it is necessary to introduce additional domain information, such as domain index.

## 5 Related Work

In this section, we review the progress of OOD prediction tasks. A research perspective for OOD prediction is from a causality viewpoint (Zhou et al., 2021; Shen et al., 2021). Based on the postulate that the causal variables are invariant and less vulnerable to distribution shifts, a bunch of methods identify the invariant causal features behind the observation data by enforcing invariance in the learning process. Different works consider causality across multiple domains in different ways. One series of research called the causal inference-based methods model the invariance across domains by causality explanation, which builds a causal graph of data generative process (Pfister et al., 2019; Rothenhäusler et al., 2018; Heinze-Deml et al., 2018; Gamella & Heinze-Deml, 2020; Oberst et al., 2021; Zhang et al., 2015). The other series of methods consider invariant learning from a causality aspect. They formulate the invariant causal mechanisms by representation rather than causal variables. Invariant risk minimization (IRM) methods (Arjovsky et al., 2019) provide a solution for learning invariant variables and functions. Under this viewpoint, some pioneering work (Ahuja et al., 2020a; Chen et al., 2022b; Krueger et al., 2021; Lu et al., 2021; Ahuja et al., 2021; Lin et al., 2022) further extend the IRM framework by considering game theory, variance penalization, information theory, nonlinear prediction functions, and some recent works apply the IRM framework to large neural networks (Jin et al., 2020; Gulrajani & Lopez-Paz, 2020). In this paper, different from the aforementioned works which consider to learn the invariant information, we think that information satisfying invariance is not enough to be most appropriate for the generalization task. We thus focus on learning to extract the most essential information from observations with a ground on the sufficient and necessary causal theorem. In the main text, we only provide a review of OOD prediction. We further elaborate on the correlation with other lines of work, such as domain adaptation, causal discovery, representation learning, causal disentanglement, and contrastive learning, in Appendix F.

## 6 Experiments

In this section, we verify the effectiveness of CaSN using synthetic and real-world OOD datasets.

### 6.1 Setups

**Synthetic data.** The effectiveness of the proposed method is demonstrated by examining whether it can learn the essential information (i.e., sufficient and necessary causes) from source data. To this

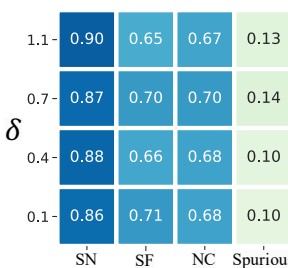  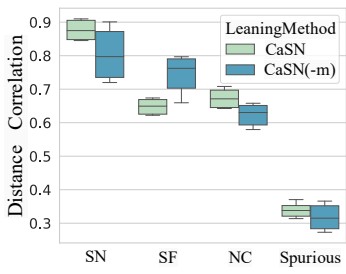

(a) Spurious degree $s = 0.1$    (b) Spurious degree $s = 0.7$    (c) Results of CaSN and the CaSN(-m)

Figure 2: The synthetic results for validating the property of learned representation under different spurious degrees in data, $s = 0.1$ for (a) and $s = 0.7$ for (b), the x-axis shows different causal information y-axis shows the choice of $\delta$. (c) The results of the feature identification when $s = 0.7$.

end, based on the causal graph in Figure 1(b).1 we designed a synthetic data generator that produced a sample set $\{\mathbf{x}_i\}_{i=1}^n$ with corresponding labels $\{y_i\}_{i=1}^n$. Four types of information were considered, including: (i) SN: Sufficient and Necessary Cause $\text{sn}_i$ of $y_i$. The value of $y_i$ is directly calculated as $y_i = \text{sn}_i \bigoplus B(0.15)$, where $\bigoplus$ represents the XOR operation and $B(0.15)$ is a Bernoulli distribution with a probability of $0.15$ to generate 1. (ii) SF: sufficient and unnecessary cause $\text{sf}_i$ of $y_i$. $\text{sf}_i$ is a transformation of $\text{sn}_i$. We set $\text{sf}_i = B(0.1)$ when $\text{sn}_i = 0$, and $\text{sf}_i = \text{sn}_i$ when $\text{sn}_i = 1$. SF is designed to decrease the probability of necessity (i.e. $P(Y = 0|\text{SN} = 0)$). (iii) NC: insufficient and necessary cause $\text{nc}_i$ of $y_i$. We set $\text{nc}_i = \text{I}(\text{sn}_i = 1) \cdot B(0.9)$. NC is designed to decrease the probability of sufficiency (i.e. $P(Y = 1|\text{SN} = 1)$). (iv) Spurious: spurious correlation information $\text{sp}_i$. Spurious correlated information is generated by $s * \text{sn}_i * \mathbf{1}_d + (1 - s)\mathcal{N}(0, 1)$, where $d$ denotes dimension and $s$ denotes the **spurious degree**. When $s$ gets higher, the spurious correlation becomes stronger in data $\mathbf{x}$. We select $d = 5$ and $s \in \{0.1, 0.7\}$. in the synthetic generative process and develop a non-linear function to generate $\mathbf{x}$ from $[\text{sn}_i, \text{sf}_i, \text{nc}_i, \text{sp}_i]$. We use Distance Correlation (Jones et al., 1995) as evaluation metrics to measure the correlation between the learned representation $\mathbf{C}$ and ground information (i.e. SN, SF, NC, SP). We provide an ablation study of CaSN without the Monotonicity evaluator CaSN(-m) in comparison results, which evaluates the effectiveness of CaSN.

**Performance on OOD prediction task.** The proposed method CaSN is implemented based on codebase DomainBed (Gulrajani & Lopez-Paz, 2020). We provide three implementations of our method, which are CaSN, CaSN(irm) and CaSN(mmd) with the same architecture but using Eq.(8), Eq.(8) $+L_{\text{irm}}$ and Eq.(8) $+L_{\text{mmd}}$ as their final objective, respectively. We compare CaSN with several common baselines, including **ERM**(Vapnik, 1999), **IRM** (Arjovsky et al., 2019), **GroupDRO** (Sagawa et al., 2019), **Mixup** (Xu et al., 2020), **MLDG** (Li et al., 2018a), **MMD** (Li et al., 2018b), **DANN** (Ganin et al., 2016) and **CDANN** (Li et al., 2018c), where the best accuracy scores are directly given by training-domain-validation in Gulrajani & Lopez-Paz (2020). We test the performance on commonly used ColoredMnist (Ahuja et al., 2020a), PACS (Li et al., 2017), and VLCS (Fang et al., 2013) datasets. During the experiment process, we adjust the hyperparameters provided by DomainBed and extra hyperparameters $\delta$ and $\lambda$ in CaSN. The results show the mean and standard error of accuracy by executing the experiments randomly 2 times on 40 randomly selected hyperparameters. We also provide the extra experiments on large-scale spurious correlation dataset SpuCo (Joshi et al., 2023). Due to the page limitation, more experiment setups and results are provided in Appendix B.

### 6.2 Learning Sufficient and Necessary Causal Representations

We conducted experiments on synthetic data to verify the effectiveness of the learned representation. In experiments, we use single domain with different degrees of spurious correlation. The experiments aimed to demonstrate the properties of the learned representation and answer the following question:

**Does CaSN capture the sufficient and necessary causes?** We present the results in Figure 2 (a) and (b), which show the distance correlation between the learned representation and four ground truths: Sufficient and Necessary cause (SN, SF, NC and Spurious). A higher distance correlation indicates a better representation. From both Figure 2 (a) (b), we found that CaSN achieves higher distance correlations with the ground truths (e.g., SN, SF, and NC) and lower correlations with spurious factors compared to other methods. As an example, we consider Figure 2 (a) with $\delta = 1.1$. We obtain distance correlations of $\{0.90, 0.65, 0.67, 0.13\}$ for SN, SF, NC, and spurious factors, respectively. We found that when we set $\delta$ as a large value $1.1$, CaSN captures more essential information SN.

Table 1: Results on PACS and VLCS dataset

| Dataset | PACS | | | | | | VLCS | | | | | |
|---|---|---|---|---|---|---|---|---|---|---|---|---|
| Algorithm | A | C | P | S | Avg | Min | C | L | S | V | Avg | Min |
| ERM | $84.7 \pm 0.4$ | $80.8 \pm 0.6$ | $97.2 \pm 0.3$ | $79.3 \pm 1.0$ | 85.5 | 79.3 | $97.7 \pm 0.4$ | $64.3 \pm 0.9$ | $73.4 \pm 0.5$ | $74.6 \pm 1.3$ | 77.5 | 64.3 |
| IRM | $84.8 \pm 1.3$ | $76.4 \pm 1.1$ | $96.7 \pm 0.6$ | $76.1 \pm 1.0$ | 83.5 | 76.4 | $98.6 \pm 0.1$ | $64.9 \pm 0.9$ | $\mathbf{73.4 \pm 0.6}$ | $\mathbf{77.3 \pm 0.9}$ | 78.5 | 64.9 |
| GroupDRO | $83.5 \pm 0.9$ | $79.1 \pm 0.6$ | $96.7 \pm 0.3$ | $78.3 \pm 2.0$ | 84.4 | 79.1 | $97.3 \pm 0.3$ | $63.4 \pm 0.9$ | $69.5 \pm 0.8$ | $76.7 \pm 0.7$ | 76.7 | 63.4 |
| Mixup | $86.1 \pm 0.5$ | $78.9 \pm 0.8$ | $\mathbf{97.6 \pm 0.1}$ | $75.8 \pm 1.8$ | 84.6 | 78.9 | $98.3 \pm 0.6$ | $64.8 \pm 1.0$ | $72.1 \pm 0.5$ | $74.3 \pm 0.8$ | 77.4 | 64.8 |
| MLDG | $86.4 \pm 0.8$ | $77.4 \pm 0.8$ | $97.3 \pm 0.4$ | $73.5 \pm 2.3$ | 83.6 | 77.4 | $97.4 \pm 0.2$ | $65.2 \pm 0.7$ | $71.0 \pm 1.4$ | $75.3 \pm 1.0$ | 77.2 | 65.2 |
| MMD | $86.1 \pm 1.4$ | $79.4 \pm 0.9$ | $96.6 \pm 0.2$ | $76.5 \pm 0.5$ | 84.6 | 79.4 | $97.7 \pm 0.1$ | $64.0 \pm 1.1$ | $72.8 \pm 0.2$ | $75.3 \pm 3.3$ | 77.5 | 64.0 |
| DANN | $86.4 \pm 0.8$ | $77.4 \pm 0.8$ | $97.3 \pm 0.4$ | $73.5 \pm 2.3$ | 83.6 | 77.4 | $\mathbf{99.0 \pm 0.3}$ | $65.1 \pm 1.4$ | $73.1 \pm 0.3$ | $77.2 \pm 0.6$ | $\mathbf{78.6}$ | 65.1 |
| CDANN | $84.6 \pm 1.8$ | $75.5 \pm 0.9$ | $96.8 \pm 0.3$ | $73.5 \pm 0.6$ | 82.6 | 75.5 | $97.1 \pm 0.3$ | $65.1 \pm 1.2$ | $70.7 \pm 0.8$ | $77.1 \pm 1.5$ | 77.5 | 65.1 |
| **CaSN (base)** | $\mathbf{87.1 \pm 0.6}$ | $80.2 \pm 0.6$ | $96.2 \pm 0.3$ | $80.4 \pm 0.2$ | $\mathbf{86.0}$ | 80.2 | $97.5 \pm 0.6$ | $64.8 \pm 1.9$ | $70.2 \pm 0.5$ | $76.4 \pm 1.7$ | 77.2 | 64.8 |
| **CaSN (irm)** | $82.1 \pm 0.3$ | $77.9 \pm 1.8$ | $93.3 \pm 0.8$ | $\mathbf{80.6 \pm 1.0}$ | 83.5 | 77.9 | $97.8 \pm 0.3$ | $65.7 \pm 0.8$ | $72.3 \pm 0.4$ | $77.0 \pm 1.4$ | 78.2 | 65.7 |
| **CaSN (mmd)** | $84.7 \pm 0.1$ | $\mathbf{81.4 \pm 1.2}$ | $95.7 \pm 0.2$ | $80.2 \pm 0.6$ | 85.5 | $\mathbf{81.4}$ | $98.2 \pm 0.7$ | $\mathbf{65.9 \pm 0.6}$ | $71.2 \pm 0.3$ | $76.9 \pm 0.7$ | 78.1 | $\mathbf{65.9}$ |

However, the result of CaSN decreases when $\delta = 0.1$, which suggests that CaSN tends to capture the most essential information when $\delta$ is set to a larger value. This phenomenon aligns with Semantic Separability. We then compare Figure 2 (a) and (b). As an example, when $\delta = 1.1$, CaSN achieves distance correlations of 0.9 and 0.91 for SN on $s = 0.1$ and $s = 0.7$, respectively. The distance correlation with spurious information is 0.13 and 0.37 for $s = 0.1$ and $s = 0.7$, respectively. The results show that when more spurious correlations are in data, CaSN tends to capture information from those spurious correlations, but the algorithm is still able to get sufficient and necessary causes.

**Ablation study.** In Figure 2(c), we provide the comparison results between CaSN and the CaSN(-m) that removes the Monotonicity measurement on synthetic data. The figure demonstrates the distance correlation recorded over 5 experiments. The green bars indicate the distance correlation between learned representation and ground truth by CaSN. CaSN can capture the desired information SN compared to others. As the blue bars show, the CaSN(-m) can better capture the causal information (e.g. SN, SF and NC) rather than spurious correlation. It can not stably identify SN, compared to SF. CaSN(-m) can be regarded as the method that only cares Exogeneity. The results support the theoretical results in Theorem 4.3, which show the effectiveness of introducing Monotonicity term.

## 6.3 Generelization to Unseen Domain

The results of the OOD generalization experiments on PACS and VLCS datasets are presented in Tables 1. Due to page limitation, we provide the results on ColoredMNIST in Table 2. The baseline method results are from Kilbertus et al. (2018). The proposed CaSN method exhibits good OOD generalization capability on both PACS and VLCS datasets. In Table 1, CaSN achieves the best average performance over 4 domains by 86.0 on PACS. On the VLCS, CaSN(irm) achieves a good average performance of 78.2, which is close to the best state-of-the-art performance achieved by DANN. For worst-domain test accuracies, the proposed method CaSN outperforms all the baseline methods. An intuitive explanation for the good performance of CaSN is that it aims to identify and extract the most essential information from observation data, excluding unnecessary or insufficient information from the optimal solution. This enables CaSN to better generalize on the worst domain.

## 7 Conclusion

In this paper, we consider the problem of learning causal representation from observation data for generalization on OOD prediction tasks. We propose a risk based on the probability of sufficient and necessary causes (Pearl, 2009), which is applicable OOD generalization tasks. The learning principle leads to practical learning algorithms for causal representation learning. Theoretical results on the computability of PNS from the source data and the generalization ability of the learned representation are presented. Experimental results demonstrate its effectiveness on OOD generalization.

## 8 Acknowledgement

We are thankful to Juzheng Miao, Xidong Feng, Kun Lin and Jingsen Zhang for their constructive suggestions and efforts on OOD generalization experiments and for offering computation resources. We also thank Congmin Ji for checking the correctness of the mathematical proof, Pengfei Zheng for his helpful discussions and the anonymous reviewers for their constructive comments on an earlier version of this paper. Furui Liu is supported by the National Key R&D Program of China (2022YFB4501500, 2022YFB4501504). Jianhong Wang is fully supported by UKRI Turing AI World-Leading Researcher Fellowship, $EP/W002973/1$.

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

# A  Discussion

In this section, we provide discussions of how to better understand PNS value, the understanding of Theorem 4.3, the connection on domain generalization, and the limitations of this work.

## A.1  Examples to Understand PNS Value

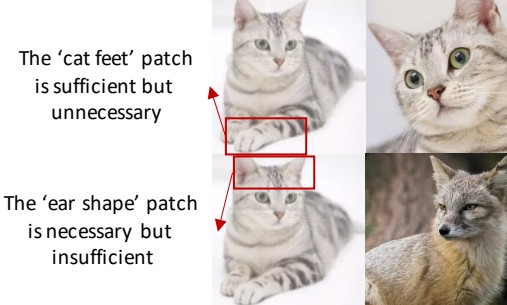

The 'cat feet' patch is sufficient but unnecessary

The 'ear shape' patch is necessary but insufficient

Figure 3: Example for causal sufficiency and necessity in image classification problem. The images on the left are for training and the rights are for OOD tests.

We provided two examples to help understand the PNS (Probability of Necessity and Sufficiency) value. We will add the following explanations to our revision.

**Example.1.** We use the feature 'has cat legs' represented by the variable $C$ (taking binary values 1 or 0) to predict the label of being a cat or a fox. 'has cat legs' is the sufficient but unnecessary cause because the image contains cat legs must have cat but cat image might not contain cat leg.

We assume $P(Y_{do(C=1)} = 1) = 1$ and $P(Y_{do(C=0)} = 0) = 0.5$, $P(Y = 1) = 0.75$, $P(C = 1, Y = 1) = 0.5$, $P(C = 0, Y = 0) = 0.25$, $P(C = 0, Y = 1) = 0.25$.

Now, applying the concept of the probability of sufficiency and necessity, we obtain:

Probability of necessity: $P(Y_{do(C=0)} = 0 | Y = 1, C = 1) = \frac{P(Y=1) - P(Y_{do(C=0)}=1)}{P(Y=1, C=1)} = \frac{0.75 - 0.5}{P(Y=1, C=1)} = 0.5$

Probability of sufficiency: $P(Y_{do(\mathbf{C}=1)} = 1 | Y = 0, C = 0) = \frac{P(Y_{do(C=1)}=1) - P(Y=1)}{P(Y=0, C=0)} = \frac{1 - 0.75}{P(Y=1, C=1)} = 1$

In this example, we can state that variable $C$ has a probability of being a sufficient cause. Note that the calculation of PN and PS are from eq.9.29 and eq.9.30 in Chapter 9 at (Pearl, 2009).

**Example.2.** If we use feature 'pointy ear' $C$ (taking values 1, 0. Value 1 means pointy ear), to predict $Y$ (cat 1, other animal 0). Since a cat must have pointy ears but if an animal has pointy ears, it should not be a cat, we assume $P(Y_{do(C=1)} = 1) = 0.5$ and $P(Y_{do(C=0)} = 0) = 1$, $P(Y = 1) = 0.25$, $P(C = 1, Y = 1) = 0.25$, $P(C = 0, Y = 0) = 0.5$, $P(C = 0, Y = 1) = 0.25$.

Now, applying the concept of the probability of sufficiency and necessity, we obtain:

Probability of necessity: $P(Y_{do(C=0)} = 0 | Y = 1, X = 1) = 1$

Probability of sufficiency: $P(Y_{do(C=1)} = 1 | Y = 0, X = 0) = 0.5$

In this example, we can state that variable $C$ has a probability of being a necessary cause.

**Example.3.** If we use feature 'eye size' $\mathbf{C}$ (taking values 1, 0.5 or 0 with the probability of $\frac{1}{3}$, respectively, the lower value means smaller eye size on face), to predict $Y$ (cat 1 or fox 0). Assuming $P(Y_{do(C=1)} = 1) = 1$, $P(Y_{do(C=0.5)} = 1) = 0.5$ and $P(Y_{do(C=0)} = 1) = 0$ (cat have larger eye size than fox). In this example, we have $P(C = 0, Y = 0) = P(C = 1, Y = 1) = \frac{1}{3}$ and $P(C = 0.5, Y = 0) = P(C = 0.5, Y = 1) = \frac{1}{6}$.

In Definition 2.1, $\text{PNS}(c, \bar{c})$ can take multiple choice. We provide Case.1: $c = 1$ and $\bar{c} = 0.5$ and Case.2: $c = 1$ and $\bar{c} = 0$.

Case.1 In this case, $\text{PN} = P(Y_{do(C=0.5)} = 0 | C = 1, Y = 1) = 0$

$\text{PS} = P(Y_{do(C=1)} = 1 | Y = 0, C = 0.5) = 3$

$\text{PNS}(1, 0.5) = \text{PN}P(C = 1, Y = 1) + \text{PS}P(Y = 0, C = 0.5) = 0.5$

Case.2 In this case, $\text{PN} = P(Y_{do(C=0)} = 0 | C = 1, Y = 1) = 1.5$

$\text{PS} = P(Y_{do(C=1)} = 1 | Y = 0, C = 0) = 1.5$

$\text{PNS}(1, 0) = P(C = 1, Y = 1) + \text{PS}P(Y = 0, C = 0) = 1.$

We note that Case.1 indicate the feature $C$ has more sufficiency than necessity. Thus, if $C$ is not binary, we should consider the worst-case PNS value.

We introduced a PNS risk to investigate the PNS value between the representation and Y. When the algorithm optimizes the objective function to minimize the PNS risk, it indicates that the representation we found contains more sufficient and necessary causal information.

Considering there are multiple choices of $c$ and $\bar{c}$ (like Example.2), we extend our algorithm by min-max optimization (Eq. 8 in the main text), which aims to find the worst $\bar{c}$ for $c$ with highest PNS risk.

## A.2 Causal Assumptions and Invariant Learning

Our assumption of the causal graph in Figure 1 is based on previous works on out-of-distribution (OOD) generalization, such as the causal graph in (Liu et al., 2021a; Ahuja et al., 2020b, 2021), they conclude the assumptions in OOD generalization task as (1) fully informative invariant features (FIIF): Feature contains all the information about the label that is contained in input. (2) Non-spurious: The environment information is not correlated with the causal information. (2) partially informative invariant features (PIIF). Our assumption $\mathbf{X} \perp Y \mid \mathbf{C}$ holds in the case of FIIF. As presented in Ahuja et al. (2021) page 3, IRM is based on the assumption of PIIF, where $\mathbf{X} \perp Y \mid \mathbf{C}$ does not hold and IRM fails in the case of FIIF. These three causal assumptions are shown in Figure 1 (b).

Given that in the domain generalization problem, the target domain is unavailable, $\beta_k$ and $\eta$ cannot be optimized directly. Thus, instead of directly optimizing the term $\beta_k$, we address the OOD generalization problem based on theoretical results in Theorem 4.3, which shows that the algorithm can identify the invariant features. We will then take the invariant learning under Assumption 4.2(1) as an example to show the relationship between invariant learning and satisfaction of Exogeneity. Theorem 4.3 establishes the equivalence between learning to optimize the Eq. (8) and intuitively demonstrates that by optimizing the objective function, the method learns representation $\mathbf{C}$ that satisfies conditional independence. Specifically, the logic is as follows:

i) In Section 4.3, we describe that our paper is based on the assumption of conditional independence $\mathbf{X} \perp Y | \mathbf{C}$ (PIIF assumption in Ahuja et al. (2021)), where $C$ is the invariant variable.

ii) Theorem 4.3 describes that the proposed algorithm makes representation satisfy such conditional independence. The proof of Theorem 4.3 indicates the learned representation is minimal sufficient statistics of $Y$ (Definition E.2). Therefore, the learned representation identifies the essential information relative to $Y$ in cause $\mathbf{C}$ theoretically.

Combining the above two points, the conclusion of Theorem 4.3 implies that the invariant variable can be learned by objective function Eq. (8).

## A.3 Framework on Domain Adaptation

In the main text, we provide the learning framework by PNS on OOD generalization task with the practical algorithm. Compared with OOD generalization, Domain adaptation is a task with the data from test distributions provided in the learning process. The framework can be extended to the Domain Adaptation task, extra samples from the test domain help the method to get better generalization performance. Same with the learning framework for OOD generalization, we start from Proposition 3.1. The Eq. (5) is adoptable in the domain adaptation tasks. In that case, the

observation data $\mathbf{x}$ in the test domain is provided during the source process. Therefore, the terms in Eq. (5) which do not require the label information $Y$ can be directly evaluated on the test domain. The same with OOD Generalization, for the terms which need to be calculated from label information, we link the test risk $R_t$ and source risk $R_s$ by the divergent measurement $\beta_k(\mathcal{T}\|\mathcal{S})$. The divergence between $\mathcal{T}$ and $\mathcal{S}$ in infinite norm measures support of the source domain that is included in the test domain. We define $t\backslash s$ as the distribution of the part of test domain distribution $\mathcal{T}$ which is not included in $\mathcal{S}$, we give the upper bound of the worst risk below.

$$\eta_{t\backslash s}(\mathbf{X}, Y) := P_t(\mathbf{X} \times Y \notin \operatorname{supp}(\mathcal{S})) \cdot \sup R_{t\backslash s}(\mathbf{w}, \phi, \xi).$$

The value of $\eta_s$ is always lower than $P_{(\mathbf{x},y,)\sim\mathcal{T}}((x,y) \notin \operatorname{supp}(\mathrm{t}))$, which indicates the data distribution for domain $t$ not covered by domain $s$.

**Theorem A.1.** *Given a source domain $\mathcal{S}$ and a test domain $\mathcal{T}$, when $\mathcal{T}(\mathbf{x})$ is provided then the risk on the test domain is bounded by the risk on source domain as follows.*

$$R_t(\mathbf{w}, \phi, \xi) \leq \lim_{k \to +\infty} M_t^{\mathbf{w}}(\phi, \xi) + \beta_k(\mathcal{T}\|\mathcal{S})(2[SF_s(\mathbf{w}, \phi)]^{1-\frac{1}{k}}) + \eta_{t\backslash s}(\mathbf{X}, Y). \tag{9}$$

Theorem A.1 shows the face of our proposed framework under a domain adaptation perspective. In domain adaptation task, we can evaluate the are not available in real-world scenarios, we only have the data samples from the test domain. The proof of Theorem A.1 is similar to Theorem 3.2, the difference is that in domain adaptation, the term $M_t^{\mathbf{w}}(\phi, \xi)$ can be directly estimated on the provided test data. Because this term does not require the label information.

## A.4 Limitations

Firstly, we propose a risk satisfying the identification condition Exogeneity under the assumption of a common causal graph in OOD generalization. However, there exist corner cases of causal assumption like anti-causal and confounder, etc. Hence, we plan to extend our framework with more causal assumptions. In addition, the theorem in our paper is based on the linear classification function. Future work would look into more complex functions in non-linear settings. We leave this exciting direction for future work as it is out of the scope of this paper.

## B  Implementation Details

In this section, we introduce the experimental setups and implementation details include the generate rules of synthetic dataset, real-world benchmarks, evaluation strategies, model architecture and hyperparameters setting.

### B.1  Synthetic Dataset

In order to evaluate whether the proposed method captures the most essential information sufficient and necessary causes or not, we design synthetic data which contains four part of variables:

- **Sufficient and necessary cause** (SN): the sufficient and necessary cause is generated from a Bernoulli distribution $\mathrm{sn}_i \sim B(0.5)$, and the label $y_i$ is generated based on the sufficient and necessary cause where $y_i = \mathrm{sn}_i \bigoplus B(0.15)$. Since $Y$ is generated from SN $\in \{0, 1\}$. The sufficient and necessary cause has high probability of $P(Y = 0|do(\mathrm{SN} = 0)) + P(Y = 1|do(\mathrm{SN} = 1))$.

- **Sufficient and unnecessary cause** (SF): From the definition and identifiability results of PNS, the sufficient and unnecessary cause SF $\in \{0, 1\}$ in synthetic data has the same probability with SN, i.e. $P(Y = 1|do(\mathrm{SF} = 1)) = P(Y = 1|do(\mathrm{SN} = 1))$, but has lower the probability of $P(Y = 0|do(\mathrm{SF} = 0))$ than $P(Y = 0|do(\mathrm{SN} = 0))$. To generate the value of $\mathrm{sf}_i$, we design a transformation function $f_{\mathrm{sf}} : \{0, 1\} \to \{0, 1\}$ to generate $\mathrm{sf}_i$ from sufficient and necessary cause value $\mathrm{sn}_i$.

$$\mathrm{sf}_i = f_{\mathrm{sf}}(\mathrm{sn}_i) := \begin{cases} B(0.1) & \text{where } \mathrm{sn}_i = 0 \\ \mathrm{sn}_i & \text{where } \mathrm{sn}_i = 1 \end{cases}. \tag{10}$$

From the functional intervention defined by (Puli et al., 2020; Pearl, 2009; Wang & Jordan, 2021), we demonstrate the identifiability property to connect the SF and $Y$ in following lemma:

**Lemma B.1.** *Let SF := $f_{sf}(SN)$, the intervention distribution $P(Y|do(SF = sf_i))$ is identified by the condition distribution $P(Y|SF = sf_i)$.*

**proof**:

$$P(Y|do(\text{SF} = \text{sf}_i)) = \int_{\text{SN}} p(y|do(\text{SN}))p(\text{SN}|f_{\text{sf}}(\text{SN}) = \text{sf}_i)d\,\text{SN}$$
$$= \int_{\text{SN}} p(y|\text{SN})p(\text{SN}|f_{\text{sf}}(\text{SN}) = \text{sf}_i)d\,\text{SN}. \tag{11}$$

The lemma shows that even $Y$ is only generated from SN in our synthetic simulator, the sufficient cause SF is exogenous relative to $Y$.

- **Insufficient and necessary cause** (NC): From the definition and identifiability results of PNS, the insufficient and necessary cause has the same probability of $P(Y = 0|do(\text{NC} = 0))$ with $P(Y = 0|do(\text{SN} = 0))$ but lower the probability of $P(Y = 1|do(\text{NC} = 1))$ than $P(Y = 1|do(\text{SN} = 1))$. To generate the value of NC, we design a transformation function $f_{\text{nc}} : \{0, 1\} \to \{0, 1\}$ to generate $\text{nc}_i$ from sufficient and necessary cause $\text{sn}_i$. The generating process of $\text{nc}_i$ is defined below, and $\text{nc}_i$ is the cause of $y$. Similar with identifiability result of SF in Lemma B.1, NC is exogenous relative to $Y$.

$$\text{nc}_i = f_{\text{nc}}(\text{sn}_i) := \text{sn}_i * B(0.9) \tag{12}$$

- **Spurious**: We also generate the variable with spurious corrlated with sufficient and necessary cause. We set the degree of spurious correlation as $s$, the generator is defind as $\text{sp}_i = s * \text{sn}_i * \mathbf{1}_d + (1 - s)\mathcal{N}(0, 1)$, where $\mathcal{N}(0, 1)$ denotes the Gaussian distribution, we select $d = 5$ in synthetic generative process. When $s$ gets higher, the spurious correlation becomes stronger in the data sample. We select $s = 0.1$ and $s = 0.7$ in synthetic experiments.

To make the data more complex, we apply a non-linear function to generate $\mathbf{x}$ from $\{\text{sn}_i, SF_i, \text{nc}_i, \text{sp}_i\}$. We fistly generate a temporary vector, which is defind as $\mathbf{t} = [\text{sn}_i * \mathbf{1}_d, SF_i * \mathbf{1}_d, \text{nc}_i * \mathbf{1}_d, \text{sp}_i * \mathbf{1}_d] + \mathcal{N}(0, 0.3)$. Then we define functions $\kappa_1(\mathbf{t}) = \mathbf{t} - 0.5$ if $\mathbf{t} > 0$, otherwise $\kappa_1(\mathbf{t}) = 0$ and $\kappa_2(\mathbf{t}) = \mathbf{t} + 0.5$ if $\mathbf{t} < 0$, otherwise $\kappa_2(\mathbf{t}) = 0$. $\mathbf{x}$ is generated by $\mathbf{x} = \sigma(\kappa_1(\mathbf{t}) \cdot \kappa_1(\mathbf{t}))$, where $\sigma : \mathbb{R}^{4d} \to \mathbb{R}^{4d}$ is sigmoid function. We generate 20000 data samples for training and 500 data samples for evaluation.

## B.2  Experiment setup on synthetic data

**Methods.** In the synthetic experiment, we compare the proposed CaSN with the reduced method CaSN (-m). In the CaSN(-m), we remove the monotonicity evaluator $M_s^{\mathbf{w}}(\phi, \xi)$ and the $\delta$'s constraint $\|\mathbf{c} - \bar{\mathbf{c}}\|_2 \geq \delta$ from the objective Eq. (8).

**Evaluation.** We evaluate the distance correlation (Jones et al., 1995) between the learned representation $\hat{\mathbf{c}}_i$ and four features $\{\text{sn}_i, \text{sf}_i, \text{nc}_i, \text{sp}_i\}$. For example, the larger distance correlation value between $\hat{\mathbf{c}}_i$ and $\text{sn}_i$ means $\hat{\mathbf{c}}_i$ contains more information of $\text{sn}_i$.

**Model Architecture.** For the synthetic dataset, we use the three-layer MLP networks with an activate function as the feature inference model. Denoting $V_k$ as weighting parameters, the neural network is designed as $V_1\text{ELU}(V_2\text{ELU}(V_3[\mathbf{x}]))$, where $ELU()$ is the activation function (Clevert et al., 2015). The dimension of hidden vectors calculated from $V_1, V_2, V_3$ are specified as $64, 32, 128$ separately. Since the output of the feature learning network is consist of the mean and variance vector, the dimension of representation is $64$. The prediction neural networks $\mathbf{w}$ which predict $y$ by representation $\mathbf{c}$.

## B.3  Experiment setup on DomainBed

**Dataset.** In addition to the experiments on synthetic data, we also evaluate the proposed method on real-world OOD generalization datasets. We use the code from the popular repository DomainBed (Gulrajani & Lopez-Paz, 2020), and use the dataset which is downloaded or generated by the DomainBed code. The dataset includes the following three:

- **PACS** (Li et al., 2017): PACS dataset contains overall $9,991$ data samples with dimension $(3, 224, 224)$. There are 7 classes in the dataset. The datasets have four domains *art, cartoons, photos, sketches*.

- **VLCS** (Fang et al., 2013): VLCS dataset contains overall 10, 729 data samples with dimension (3, 224, 224). There are 5 classes in the dataset. The datasets have four domains *Caltech101, LabelMe, SUN09, VOC2007*.

The DomainBed randomly split the whole dataset as training, validation and test dataset.

**Baselines.** For OOD Generalization task, we consider following baselines realized by DomainBed.

- **Empirical Risk Minimization (ERM)** (Vapnik, 1999) (ERM, Vapnik [1998]) is a common baseline that minimizes the risk on all the domain data.

- **Invariant Risk Minimization (IRM)** (Arjovsky et al., 2019) minimize the the risk for different domains separately. It extract the representation on all the domains and project them on a shared representation space. Then, all the domain share one prediction model. The algorithm is formed as bi-level optimization objective.

- **Group Distributionally Robust Optimization (DRO)** (Sagawa et al., 2019) weigh the different domains by their evaluated error. It reweights the optimization of domains in objective. The domain with a larger error gets a larger weight.

- **Inter-domain Mixup (Mixup)** (Xu et al., 2020; Yan et al., 2020; Wang et al., 2020), ERM is performed for linear interpolation of samples of random pairs from domains and their labels.

- Meta-Learning for Domain Generalization (MLDG) (Li et al., 2018a) uses meta learning strategy (Finn et al., 2017) to learn the cross domain generalization.

- **MMD** method for domain generalization (Li et al., 2018b) method uses the MMD (Gretton et al., 2012) of the feature distribution to measure the distance of domain representation. They learn the invariant representation across domains by minimize MMD.

- **Domain-Adversarial Neural Networks DANN** (Ganin et al., 2016) uses adversarial networks to learn the invariant features across domains.

- **Class-conditional DANN CDANN** (Li et al., 2018c) is an extension of DANN, which is learned to match the conditional distribution by giving the label information.

**Evaluation.** All the evaluation results are reported from the DomainBed paper (Gulrajani & Lopez-Paz, 2020). We use the evaluation strategy from DomainBed. DomainBed provides the evaluation strategy Training-domain validation set. The training dataset is randomly split as training and validation datasets, the hyperparameters are selected on the validation dataset, which maximizes the performance of the validation dataset. The overall accuracy results are evaluated on the test dataset rather than the validation dataset.

All the experiments are conducted based on a server with a 16-core CPU, 128g memory and RTX 5000 GPU.

**Implementation Details.** We implement CaSN based on the **DomainBed** repository. The feature extractor and classifier use different neural networks across different datasets.

We set distribution of representation $\mathbf{C}$ as $\hat{P}_s^\phi(\mathbf{C}|\mathbf{X}=\mathbf{x}) = \mathcal{N}(\mu(\mathbf{x};\phi), \sigma(\mathbf{x};\phi))$ and its prior as $\pi_\mathbf{C} = \mathcal{N}(\mu_\mathbf{0}, \sigma_\mathbf{0})$. $\mu_0$ and $\sigma_0$ are pre-defined and not updated in the learning process. Similarly, for $\bar{\mathbf{C}}$, we define $\hat{P}_s^\xi(\bar{\mathbf{C}}|\mathbf{X}=\mathbf{x}) = \mathcal{N}(\bar{\mu}(\mathbf{x};\xi), \bar{\sigma}(\mathbf{x};\xi))$ and $\pi_{\bar{\mathbf{C}}} = \mathcal{N}(\bar{\mu}_\mathbf{0}, \bar{\sigma}_\mathbf{0})$. **Colored Mnist**: The model architecture on Colored Mnist dataset is given by DomainBed repository. We use MNIST ConvNet for feature extraction. The details of the layers are referred from Table 7 in Gulrajani & Lopez-Paz (2020). We simply use a linear function as the label prediction network and transformation function $T$. In the Colored Mnist dataset, the variance of $\mathbf{c}$ is fixed as $0.001$.

**PACS and VLCS**: The model architecture on PACS and VLCS datasets are both given by DomainBed repository. They use ResNet-50 for feature extraction. In DomainBed, they replace the last layer of a ResNet50 pre-trained on ImageNet and fine-tune and freeze all the bach normalization layer before fine-tune. We simply use a linear function as the label prediction network $\mathbf{w}$ and transformation function $T$. In the PACS and VLCS datasets, the variances of $\mathbf{c}$ are also fixed as $0.001$

The Hyperparameters are shown in Table 3. The general hyperparameters (e.g. ResNet, Mnist, Not Mnist) are directly given from Table 8 in Gulrajani & Lopez-Paz (2020). All the experiments run for 2 times.

Table 2: Results on Colored Mnist

| Condition | +90% | +80% | -90% | Avg | Min |
|---|---|---|---|---|---|
| ERM | $71.7 \pm 0.1$ | $72.9 \pm 0.2$ | $10.0 \pm 0.1$ | 51.5 | 10.0 |
| IRM | $72.5 \pm 0.1$ | $73.3 \pm 0.5$ | $10.2 \pm 0.3$ | 52.0 | 10.2 |
| GroupDRO | $\mathbf{73.1 \pm 0.3}$ | $73.2 \pm 0.2$ | $10.0 \pm 0.2$ | 52.1 | 10.0 |
| Mixup | $72.7 \pm 0.4$ | $73.4 \pm 0.1$ | $10.1 \pm 0.1$ | 52.1 | 10.1 |
| MLDG | $71.5 \pm 0.2$ | $73.1 \pm 0.2$ | $9.8 \pm 0.1$ | 51.5 | 9.8 |
| MMD | $71.4 \pm 0.3$ | $73.1 \pm 0.2$ | $9.9 \pm 0.3$ | 51.5 | 9.9 |
| DANN | $71.4 \pm 0.9$ | $73.1 \pm 0.1$ | $10.0 \pm 0.0$ | 51.5 | 10.0 |
| CDANN | $72.0 \pm 0.2$ | $73.0 \pm 0.2$ | $10.2 \pm 0.1$ | 51.7 | 10.2 |
| CaSN (Ours) | $72.6 \pm 0.1$ | $\mathbf{73.7 \pm 0.1}$ | $\mathbf{10.3 \pm 0.3}$ | **52.2** | **10.3** |

Table 3: Hyperparameters setting

|  | Parameter | Default value | Random distribution |
|---|---|---|---|
| Basic | batch size | 32 | $2^{\text{Uniform } (3,5.5)}$ |
|  | minimization process learning rate | 0.0001 | $10^{\text{Uniform } (-5,-3.5)}$ |
|  | maxmization process learning rate | 0.00001 | $10^{\text{Uniform } (-6,-4)}$ |
| Not Mnist | weight decay | 0 | $10^{\text{Uniform } (-6,-2)}$ |
|  | generator weight decay | 0 | $10^{\text{Uniform } (-6,-2)}$ |
| Mnist | weight decay | 0 | 0 |
|  | generator weight decay | 0 | 0 |
| CaSN | $\delta$ | 0.7 | $\{0.3, 0.7, 1.0, 1.5, 3.0\}$ |
|  | $\lambda$ | 0.01 | $\{0.001, 0.01, 0.1\}$ |
|  | Weight of constraint $\|\mathbf{c} - \bar{\mathbf{c}}\|_2$ | 0.1 | $\{0.001, 0.01, 0.1, 0.5, 0.7\}$ |
| CaSN(irm) | $\delta$ | 0.7 | $\{0.3, 0.7, 1.0, 1.5, 3.0\}$ |
|  | $\lambda$ | 0.01 | $\{0.001, 0.01, 0.1, 0.5, 0.7\}$ |
|  | Weight of constraint $\|\mathbf{c} - \bar{\mathbf{c}}\|_2$ | 0.1 | $\{0.001, 0.01, 0.1, 0.5, 0.7\}$ |
|  | Weight of IRM penalty $L_{\text{irm}}$ | 0.001 | $\{0.01, 0.001, 1e-5\}$ |
|  | IRM penalty iters | 1000 | $\{500, 1000\}$ |
| CaSN(mmd) | $\delta$ | 0.7 | $\{0.3, 0.7, 1.0, 1.5, 3.0\}$ |
|  | $\lambda$ | 0.01 | $\{0.001, 0.01, 0.1, 0.5, 0.7\}$ |
|  | Weight of constraint $\|\mathbf{c} - \bar{\mathbf{c}}\|_2$ | 0.1 | $\{0.001, 0.01, 0.1, 0.5, 0.7\}$ |
|  | Weight of MMD penalty $L_{\text{mmd}}$ | 1 | $\{1\}$ |
| Adversarial | Max optimization per iters | 500 | $\{500, 1000\}$ |

## B.4 Experiment setup on SpuCo

We provide the experiment on the large scale spurious correlation dataset SpuCo (Joshi et al., 2023). SpuCo is a Python package developed to address spurious correlations in the context of visual prediction scenarios. It includes two datasets: the simulated dataset SpuCoMNIST and the large-scale real-world dataset SpuCoANIMALS. This Python package provides code for benchmark methods on these datasets, along with test results.

**Dataset.** We test CaSN on the large-scale real-world dataset SpuCoANIMALS, which is derived from ImageNet (Russakovsky et al., 2015). This dataset captures spurious correlations in real-world settings with greater fidelity compared to existing datasets. There are 4 classes in this dataset: landbirds, waterbirds, small dog breeds and big dog breeds, which are spuriously correlated with different backgrounds. Based on the different backgrounds, this dataset contains 8 groups with more than $10,000$ samples.

**Baselines.** We realize CaSN based on the code of SpuCo. We have CaSN implementation, one is based on ERM, the other is based on the strategy of Group Balance. We compare CaSN with ERM, GroupDRO and Group Balance.

Table 4: Results on SpuCo

|  | ERM | CaSN | GroupDRO | GB | CaSN-GB |
|---|---|---|---|---|---|
| Average | 80.26±4.1 | 84.34±0.6 | 50.7±4.2 | 49.9±1.4 | 50.54±7.6 |
| Min | 8.3±1.3 | 9.7±2.7 | 36.7±0.6 | 41.1±2.7 | 42.0±0.8 |

**Hyperparameters.** The hyperparameters employed for robust retraining are detailed as follows. During the robust training phase, we set the hyperparameters as their default value:

When conducting training from scratch, we utilize SGD as the optimizer with a learning rate of 0.0001, a weight decay of 0.0001, a batch size of 128, and a momentum of 0.9. The model undergoes training for 300 epochs, with early stopping applied based on its performance on the validation set. In addition to the basic hyperparameters, we set $\delta = 3$ and $\lambda \in \{0.005, 0.01\}$.

## C  Additional Results

### C.1  OOD generalization results on Colored MNIST

Due to the page limitation in the main text, we show OOD generalization results on Colored Mnist data in Table 2.

### C.2  Domain generalization on SpuCo

The results on SpuCo (Joshi et al., 2023) based on the code SpuCo[4]. We test CaSN on SpuCoAnimal which comprises 8 different spurious correlation groups.

We set hyperparameters as $\delta = 3$ and $\lambda \in 0.005, 0.01$, while the remaining parameters followed SpuCo's default values. We conducted two types of experiments: the first involved training the model without providing any additional information, and the second involved resampling based on the spurious correlation group information, where the base models corresponded to ERM and Group Balance (GB) as presented in SpuCo. We also include GroupDRO as one of the baselines, the baseline performance is reported SpuCo. Data augmentation-based methods, ERM* and GB*, were not included in these experiments.

We reported the results in Table 4, with each result obtained from 4 fixed random seeds. The baseline results are directly taken from SpuCo. Our method showed higher average accuracy on average accuracy compared to the baseline, and even the worst spurious correlation group exhibited higher accuracy than the baseline.

## D  Proof of Theorems

In this section, we provide the proof of the risk bounds which include Proposition 3.1, Theorem 3.2 and Theorem 3.3 for OOD generalization task.

### D.1  Proposition 3.1

To explicitly form the Monotonocity evaluator, we decompose the original objective by three terms defined by sufficiency objective $SF_t(\mathbf{w}, \phi)$, necessity objective $NC_t(\mathbf{w}, \xi)$ and Monotonicity evaluator objective $M_t^{\mathbf{w}}(\phi, \xi)$. The Monotonicity $M_t^{\mathbf{w}}(\phi, \xi)$ term can be decomposed as

$$M_t^{\mathbf{w}}(\phi, \xi) = SF_t(\mathbf{w}, \phi)(1 - NC_t(\mathbf{w}, \xi)) + (1 - SF_t(\mathbf{w}, \phi))NC_t(\mathbf{w}, \xi). \tag{13}$$

The following equation understands the above decomposition.

$$\begin{aligned}
&P(\text{sign}(\mathbf{w}^\top \mathbf{c}) = \text{sign}(\mathbf{w}^\top \bar{\mathbf{c}})) \\
&= P(\text{sign}(\mathbf{w}^\top \mathbf{c}) = y)P(\text{sign}(\mathbf{w}^\top \bar{\mathbf{c}}) = y) + P(\text{sign}(\mathbf{w}^\top \mathbf{c}) \neq y)P(\text{sign}(\mathbf{w}^\top \bar{\mathbf{c}}) \neq y).
\end{aligned} \tag{14}$$

---

[4]https://github.com/BigML-CS-UCLA/SpuCo

We can further derive Eq.13 as follows.

$$
\begin{aligned}
M_t^{\mathbf{w}}(\phi,\xi) =& SF_t(\mathbf{w},\phi)(1 - NC_t(\mathbf{w},\xi)) + (1 - SF_t(\mathbf{w},\phi))NC_t(\mathbf{w},\xi) \\
=& \underbrace{SF_t(\mathbf{w},\phi) + NC_t(\mathbf{w},\xi)}_{R_t(\mathbf{w},\phi,T)} - 2SF_t(\mathbf{w},\phi)NC_t(\mathbf{w},\xi) \\
=& R_t(\mathbf{w},\phi,\xi) - 2SF_t(\mathbf{w},\phi)NC_t(\mathbf{w},\xi).
\end{aligned}
\tag{15}
$$

Then we can rewrite the original objective Eq. (4) by

$$
R_t(\mathbf{w},\phi,\xi) = M_t^{\mathbf{w}}(\phi,\xi) + 2SF_t(\mathbf{w},\phi)NC_t(\mathbf{w},\xi).
\tag{16}
$$

Then we get the results of Proposition 3.1. The reasons why we need this proposition are because the final objective need to explicitly evaluate the Monotonicity.

## D.2 Theorem 3.2

The motivation of Theorem 3.2 is to bridge the gap between the risk on the source domain and the risk on the test domain. To prove the result in Theorem 3.2, we refer to the technicals in Germain et al. (2016), We first define $o = \mathbb{E}_{(\mathbf{x},y)\sim\mathcal{T}}\mathrm{I}[(\mathbf{x},y)\notin \sup(\mathcal{S})]$, then we can get the value of $\delta$-Monotonicity measurement on the samples from test domain that are not included in source domain is:

$$
\begin{aligned}
&\mathbb{E}_{(\mathbf{x},y)\sim\mathcal{T}}\mathrm{I}[(\mathbf{x},y)\notin \mathrm{supp}(\mathcal{S})]\mathbb{E}_{\mathbf{c}\sim P_t^\phi(\mathbf{C}|\mathbf{X}=\mathbf{x})}\mathbb{E}_{\bar{\mathbf{c}}\sim P_t^\xi(\bar{\mathbf{C}}|\mathbf{X}=\mathbf{x})}\mathrm{I}[\mathrm{sign}(\mathbf{w}^\top\mathbf{c}) = \mathrm{sign}(\mathbf{w}^\top\bar{\mathbf{c}})] \\
=&o\mathbb{E}_{t\setminus s}\mathbb{E}_{\mathbf{c}\sim P_t^\phi(\mathbf{C}|\mathbf{X}=\mathbf{x})}\mathbb{E}_{\bar{\mathbf{c}}\sim P_t^\xi(\bar{\mathbf{C}}|\mathbf{X}=\mathbf{x})}\mathrm{I}[\mathrm{sign}(\mathbf{w}^\top\mathbf{c}) = \mathrm{sign}(\mathbf{w}^\top\bar{\mathbf{c}})] = oM_t^{\mathbf{w}}(\phi,\xi).
\end{aligned}
$$

Similarly, the overall risk on the samples from the test domain that is not included in the source domain is $\eta_{t\setminus s}(\mathbf{X},Y)$.

Then we change the distribution measure, we also take $M_t^{\mathbf{w}}(\phi,\xi)$ of Eq. (5) as an example.

$$
\begin{aligned}
&M_t^{\mathbf{w}}(\phi,\xi) \\
=&\mathbb{E}_{(\mathbf{x},y)\sim\mathcal{T}}\mathbb{E}_{\mathbf{c}\sim P_t^\phi(\mathbf{C}|\mathbf{X}=\mathbf{x})}\mathbb{E}_{\bar{\mathbf{c}}\sim P_t^\xi(\bar{\mathbf{C}}|\mathbf{X}=\mathbf{x})}\mathrm{I}[\mathrm{sign}(\mathbf{w}^\top\mathbf{c}) = \mathrm{sign}(\mathbf{w}^\top\bar{\mathbf{c}})] \\
=&\mathbb{E}_{(\mathbf{x},y)\sim\mathcal{S}}\frac{\mathcal{T}}{\mathcal{S}}\mathbb{E}_{\mathbf{c}\sim P_t^\phi(\mathbf{C}|\mathbf{X}=\mathbf{x})}\mathbb{E}_{\bar{\mathbf{c}}\sim P_t^\xi(\bar{\mathbf{C}}|\mathbf{X}=\mathbf{x})}\mathrm{I}[\mathrm{sign}(\mathbf{w}^\top\mathbf{c}) = \mathrm{sign}(\mathbf{w}^\top\bar{\mathbf{c}})] + oM_{t\setminus s}^{\mathbf{w}}(\phi,\xi) \\
\leq&\beta_k(\mathcal{T}\|\mathcal{S})[\mathbb{E}_{\mathbf{c}\sim P_s^\phi(\mathbf{C}|\mathbf{X}=\mathbf{x})}\mathbb{E}_{\bar{\mathbf{c}}\sim P_s^\xi(\bar{\mathbf{C}}|\mathbf{X}=\mathbf{x})}\mathrm{I}[\mathrm{sign}(\mathbf{w}^\top\mathbf{c}) = \mathrm{sign}(\mathbf{w}^\top\bar{\mathbf{c}})]^{\frac{k}{k-1}}]^{1-\frac{1}{k}} + oM_{t\setminus s}^{\mathbf{w}}(\phi,\xi) \\
=&\beta_k(\mathcal{T}\|\mathcal{S})[\mathbb{E}_{\mathbf{c}\sim P_s^\phi(\mathbf{C}|\mathbf{X}=\mathbf{x})}\mathbb{E}_{\bar{\mathbf{c}}\sim P_s^\xi(\bar{\mathbf{C}}|\mathbf{X}=\mathbf{x})}\mathrm{I}[\mathrm{sign}(\mathbf{w}^\top\mathbf{c}) = \mathrm{sign}(\mathbf{w}^\top\bar{\mathbf{c}})]]^{1-\frac{1}{k}} + oM_{t\setminus s}^{\mathbf{w}}(\phi,\xi).
\end{aligned}
\tag{17}
$$

The third line is due to Hölder inequality. For the last line, we remove the exponential term $\frac{k}{k-1}$ in $\mathrm{I}[\mathrm{sign}(\mathbf{w}^\top\mathbf{c}) = \mathrm{sign}(\mathbf{w}^\top\bar{\mathbf{c}})]^{\frac{k}{k-1}}$ since the function take the results from $\{0,1\}$. Similarly on term $SF_s(\mathbf{w},\phi)$, we can get the final bound of overall $R_t(\mathbf{w},\phi,\xi)$ as Eq. (7) shows.

## D.3 Theorem 3.3

In this theorem, we study how the risk on distribution is bounded by empirical risk. The proof of the theorem refers to popular inequality Jensen's inequality, Markov inequality and Hoeffding's inequality. We first focus on the term $SF_s(\mathbf{w},\phi)$. The sketch of the proof is that firstly we will use the variational inference process to change the measure of distribution. Then, we use Markov's inequality to calculate the bound of risk.

Define $\Delta(SF_s(\mathbf{w}, \phi)) = \widehat{SF}_s(\mathbf{w}, \phi) - SF_s(\mathbf{w}, \phi)$. We apply the variational inference trick and use Jensen's inequality to get the following inequality:

$$
\begin{aligned}
&\Delta(SF_s(\mathbf{w}, \phi)) \\
&= \widehat{SF}_s(\mathbf{w}, \phi) - SF_s(\mathbf{w}, \phi) \\
&\leq \mathbb{E}_{\mathcal{S}^n} \mathrm{KL}(\hat{P}_s^\phi(\mathbf{C}|\mathbf{X} = \mathbf{x})\|\pi_{\mathbf{C}}) - 2(n-1)\mathbb{E}_{\mathcal{S}}\mathbb{E}_{P_s^\phi(\mathbf{C}|\mathbf{X}=\mathbf{x})} \ln \frac{P_s^\phi(\mathbf{C}|\mathbf{X} = \mathbf{x})}{\pi_{\mathbf{C}}} \\
&\quad + \ln \mathbb{E}_{\mathbf{c}\sim\pi_{\mathbf{C}}} \exp(\mathbb{E}_{\mathcal{S}^n} \mathrm{I}[\mathrm{sign}(\mathbf{w}^\top\mathbf{c}) \neq y]) \\
&\quad - \ln \mathbb{E}_{\mathbf{c}\sim\pi_{\mathbf{C}}} \exp\left(\mathbb{E}_{\mathcal{S}} \mathrm{I}[\mathrm{sign}(\mathbf{w}^\top\mathbf{c}) \neq y]\right), \\
&= \mathbb{E}_{\mathcal{S}^n} \mathrm{KL}(\hat{P}_s^\phi(\mathbf{C}|\mathbf{X} = \mathbf{x})\|\pi_{\mathbf{C}}) - \mathbb{E}_{\mathcal{S}}\mathbb{E}_{P_s^\phi(\mathbf{C}|\mathbf{X}=\mathbf{x})} \ln \frac{P_s^\phi(\mathbf{C}|\mathbf{X} = \mathbf{x})}{\pi_{\mathbf{C}}} \\
&\quad + \ln \mathbb{E}_{\mathbf{c}\sim\pi_{\mathbf{C}}} \exp\left(\Delta\left(S\right)\right), \\
&= \mathbb{E}_{\mathcal{S}^n} \mathrm{KL}(\hat{P}_s^\phi(\mathbf{C}|\mathbf{X} = \mathbf{x})\|\pi_{\mathbf{C}}) - \mathbb{E}_{\mathcal{S}} \mathrm{KL}(P_s^\phi(\mathbf{C}|\mathbf{X} = \mathbf{x})\|\pi_{\mathbf{C}}) \\
&\quad + \ln \mathbb{E}_{\mathbf{c}\sim\pi_{\mathbf{C}}} \exp\left(\Delta\left(S\right)\right),
\end{aligned}
\tag{18}
$$

where $\Delta\left(S\right) = |\mathbb{E}_{\mathcal{S}^n} \mathrm{I}[\mathrm{sign}(\mathbf{w}^\top\mathbf{c}) \neq y] - \mathbb{E}_{\mathcal{S}} \mathrm{I}[\mathrm{sign}(\mathbf{w}^\top\mathbf{c}) \neq y]|$. Recall that Hoeffding's inequality, we get the following inequality.

$$
P[\Delta(S) \geq \eta] \leq \exp(-2n)\eta^2 \tag{19}
$$

Then, denoting the density function of $\Delta(S)$ as $f(\Delta(S))$

$$
\begin{aligned}
P[\Delta(S) \geq \eta] &= e^{-2n\eta^2} \\
\Rightarrow \int_\eta^\infty f(\Delta(S)) d\Delta(S) &= e^{-2n\eta^2} \\
\Rightarrow f(\eta) &= 4n\eta e^{-2n\eta^2}.
\end{aligned}
$$

Then, we get

$$
\begin{aligned}
&\mathbb{E}_{\mathcal{S}}[\exp(2(n-1)\Delta^2(S))] \\
&= \int_0^1 f(\Delta(S)) \exp(2(n-1)\Delta^2(S)) d\Delta(S) \\
&\leq \int_0^1 4n\Delta(S) \exp(-2n\Delta^2(S)) \exp(2(n-1)\Delta^2(S)) d\Delta(S) \\
&= -n \exp\left(-2\Delta^2(S)\right)|_0^1 \\
&= (1 - e^{-2})n < n
\end{aligned}
\tag{20}
$$

Combining Eq. (20) with Eq. (18). Suppose $g(\Delta(S)) = \ln \mathbb{E}_{\pi_{\mathbf{C}}}[\exp(2(n-1)\Delta^2(S))]$, since we have,

$$
\mathbb{E}_{\mathcal{S}}\mathbb{E}_{\pi_{\mathbf{C}}}[\exp\left(2(n-1)\Delta^2\left(S\right)\right)] \leq n
$$

by Markov's inequality, we further get,

$$
P_{(\mathcal{S})}[g(\Delta(S)) \geq \eta] \leq \frac{n}{e^\eta}, \tag{21}
$$

Suppose that $\eta = \ln(n/\epsilon)$, and with the probability of at least $1 - \epsilon$, we have that for all $\pi_{\mathbf{C}}$,

$$
\Rightarrow 2(n-1) \ln \mathbb{E}_{\pi_{\mathbf{C}}}[\exp\left(\Delta^2\left(S\right)\right)] \leq \eta = \ln \frac{n}{\epsilon} \tag{22}
$$

Since $\Delta S \in [0, 1]$, then we have,

$$
\exp(\Delta^2(S)) \leq (\exp(\Delta(S)))^2 \leq e \cdot \exp(\Delta^2(S)) \tag{23}
$$

The last inequality due to $\exp(\Delta^2(S)) - (\exp(\Delta(S)))^2$ is a monotonically decreasing when $\Delta S \in [0, 1]$. Then, according to Jensen's inequality, we conclude that

$$\ln(\mathbb{E}_{\mathbf{c}\sim\pi(\mathbf{C})} \exp \Delta(S))^2 \leq \ln \mathbb{E}_{\mathbf{c}\sim\pi(\mathbf{C})}(\exp \Delta(S))^2$$

$$\leq \ln(e \cdot \mathbb{E}_{\mathbf{c}\sim\pi(\mathbf{C})} \exp \Delta^2(S)) \leq \frac{1}{2(n-1)} \ln \frac{n}{\epsilon} + 1 \qquad (24)$$

$$\Rightarrow \ln(\mathbb{E}_{\mathbf{c}\sim\pi(\mathbf{C})} \exp \Delta(S)) \leq \frac{1}{4(n-1)} \ln \frac{n}{\epsilon} + \frac{1}{2}$$

Then,

$$|\widehat{SF}_s(\mathbf{w}, \phi) - SF_s(\mathbf{w}, \phi)|$$

$$\leq |\mathbb{E}_{\mathcal{S}^n} \mathrm{KL}(\hat{P}_s^\phi(\mathbf{C}|\mathbf{X}=\mathbf{x})\|\pi_{\mathbf{C}}) - \mathbb{E}_{\mathcal{S}} \mathrm{KL}(P_s^\phi(\mathbf{C}|\mathbf{X}=\mathbf{x})\|\pi_{\mathbf{C}}) + \frac{1}{4(n-1)} \ln(n/\epsilon) + \frac{1}{2}| \qquad (25)$$

$$\leq |\mathbb{E}_{\mathcal{S}^n} \mathrm{KL}(\hat{P}_s^\phi(\mathbf{C}|\mathbf{X}=\mathbf{x})\|\pi_{\mathbf{C}}) + \mathbb{E}_{\mathcal{S}} \mathrm{KL}(P_s^\phi(\mathbf{C}|\mathbf{X}=\mathbf{x})\|\pi_{\mathbf{C}}) + \frac{1}{4(n-1)} \ln(n/\epsilon) + \frac{1}{2}|.$$

According to the assumption, we have

$$|\widehat{SF}_s(\mathbf{w}, \phi) - SF_s(\mathbf{w}, \phi)| \leq \mathbb{E}_{\mathcal{S}^n} \mathrm{KL}(\hat{P}_s^\phi(\mathbf{C}|\mathbf{X}=\mathbf{x})\|\pi_{\mathbf{C}}) + \frac{1}{4(n-1)} \ln(n/\epsilon) + C. \qquad (26)$$

We get the results demonstrated in Theorem 3.3 (1).

For the term $M_s^{\mathbf{w}}(\phi, \xi)$, we define $\Delta(M_s) = M_s^{\mathbf{w}}(\phi, \xi) - \widehat{M}_s^{\mathbf{w}}(\phi, \xi)$, where $\widehat{M}_s^{\mathbf{w}}(\phi, \xi) := \mathbb{E}_{\mathcal{S}^n} \mathbb{E}_{\mathbf{c}\sim\hat{P}_s^\phi(\mathbf{C}|\mathbf{X}=\mathbf{x})} \mathbb{E}_{\bar{\mathbf{c}}\sim\hat{P}_s^\xi(\bar{\mathbf{C}}|\mathbf{X}=\mathbf{x})} \mathrm{I}[\mathrm{sign}(\mathbf{w}^\top\mathbf{c}) = \mathrm{sign}(\mathbf{w}^\top\bar{\mathbf{c}})]$. Different from Theorem 3.3 (1), the monotonicity measurement has an extra expectation on $\bar{\mathbf{c}}$. We apply Jensen's inequality again and then use the variational inference trick to get the derivation results.

$$4(n-1)^2 \Delta(M_s)^2 = 4(n-1)^2 \left(M_s^{\mathbf{w}}(\phi, \xi) - \widehat{M}_s^{\mathbf{w}}(\phi, \xi)\right)^2 \qquad (27)$$

Then, we consider the term $M_s^{\mathbf{w}}(\phi, \xi)$ and $\widehat{M}_s^{\mathbf{w}}(\phi, \xi)$ separately.

$$M_s^{\mathbf{w}}(\phi, \xi) = \mathbb{E}_{\mathcal{S}} \mathbb{E}_{\mathbf{c}\sim P_s^\phi(\mathbf{C}|\mathbf{X}=\mathbf{x})} \mathbb{E}_{\bar{\mathbf{c}}\sim P_s^\xi(\bar{\mathbf{C}}|\mathbf{X}=\mathbf{x})} \mathrm{I}[\mathrm{sign}(\mathbf{w}^\top\mathbf{c}) = \mathrm{sign}(\mathbf{w}^\top\bar{\mathbf{c}})]$$

$$= \mathbb{E}_{\mathcal{S}}[\mathbb{E}_{\mathbf{c}} \mathbb{E}_{\bar{\mathbf{c}}} \ln \frac{P_s^\phi(\mathbf{C}|\mathbf{X}=\mathbf{x})}{\pi_{\mathbf{C}}} + \mathbb{E}_{\mathbf{c}} \mathbb{E}_{\bar{\mathbf{c}}} \ln \frac{P_s^\xi(\bar{\mathbf{C}}|\mathbf{X}=\mathbf{x})}{\pi_{\bar{\mathbf{C}}}}$$

$$+ \mathbb{E}_{\mathbf{c}} \mathbb{E}_{\bar{\mathbf{c}}} \ln \frac{\pi_{\mathbf{C}}}{P_s^\phi(\mathbf{C}|\mathbf{X}=\mathbf{x})} \frac{\pi_{\bar{\mathbf{C}}}}{P_s^\xi(\bar{\mathbf{C}}|\mathbf{X}=\mathbf{x})} \exp(\mathrm{I}[\mathrm{sign}(\mathbf{w}^\top\mathbf{c}) = \mathrm{sign}(\mathbf{w}^\top\bar{\mathbf{c}})])] \qquad (28)$$

$$\leq \mathbb{E}_{(\mathbf{x},y)\sim\mathcal{S}}[\mathrm{KL}(P_s^\phi(\mathbf{C}|\mathbf{X}=\mathbf{x})\|\pi_{\mathbf{C}}) + \mathrm{KL}(P_s^\xi(\bar{\mathbf{C}}|\mathbf{X}=\mathbf{x})\|\pi_{\bar{\mathbf{C}}})]$$

$$+ \ln \mathbb{E}_{\mathbf{c}\sim\pi_{\mathbf{C}}} \mathbb{E}_{\bar{\mathbf{c}}\sim\pi_{\bar{\mathbf{C}}}} \exp(\mathbb{E}_{\mathcal{S}} \mathrm{I}[\mathrm{sign}(\mathbf{w}^\top\mathbf{c}) = \mathrm{sign}(\mathbf{w}^\top\bar{\mathbf{c}})]).$$

Similarly, for the empirical risk $\widehat{M}_s^{\mathbf{w}}(\phi, \xi)$

$$\widehat{M}_s^{\mathbf{w}}(\phi, \xi) = \mathbb{E}_{(\mathbf{x},y)\sim\mathcal{S}^n} \mathbb{E}_{\mathbf{c}\sim\hat{P}_s^\phi(\mathbf{C}|\mathbf{X}=\mathbf{x})} \mathbb{E}_{\bar{\mathbf{c}}\sim\hat{P}_s^\xi(\bar{\mathbf{C}}|\mathbf{X}=\mathbf{x})} \mathrm{I}[\mathrm{sign}(\mathbf{w}^\top\mathbf{c}) = \mathrm{sign}(\mathbf{w}^\top\bar{\mathbf{c}})]$$

$$= \mathbb{E}_{\mathcal{S}^n}[\mathbb{E}_{\mathbf{c}} \mathbb{E}_{\bar{\mathbf{c}}} \ln \frac{\hat{P}_s^\phi(\mathbf{C}|\mathbf{X}=\mathbf{x})}{\pi_{\mathbf{C}}} + \mathbb{E}_{\mathbf{c}} \mathbb{E}_{\bar{\mathbf{c}}} \ln \frac{\hat{P}_s^\xi(\bar{\mathbf{C}}|\mathbf{X}=\mathbf{x})}{\pi_{\bar{\mathbf{C}}}}$$

$$+ \mathbb{E}_{\mathbf{c}} \mathbb{E}_{\bar{\mathbf{c}}} \ln \frac{\pi_{\mathbf{C}}}{\hat{P}_s^\phi(\mathbf{C}|\mathbf{X}=\mathbf{x})} \frac{\pi_{\bar{\mathbf{C}}}}{\hat{P}_s^\xi(\bar{\mathbf{C}}|\mathbf{X}=\mathbf{x})} \exp(\mathrm{I}[\mathrm{sign}(\mathbf{w}^\top\mathbf{c}) = \mathrm{sign}(\mathbf{w}^\top\bar{\mathbf{c}})])] \qquad (29)$$

$$\leq \mathbb{E}_{(\mathbf{x},y)\sim\mathcal{S}^n}[\mathrm{KL}(\hat{P}_s^\phi(\mathbf{C}|\mathbf{X}=\mathbf{x})\|\pi_{\mathbf{C}}) + \mathrm{KL}(\hat{P}_s^\xi(\bar{\mathbf{C}}|\mathbf{X}=\mathbf{x})\|\pi_{\bar{\mathbf{C}}})]$$

$$+ \ln \mathbb{E}_{\mathbf{c}\sim\pi_{\mathbf{C}}} \mathbb{E}_{\bar{\mathbf{c}}\sim\pi_{\bar{\mathbf{C}}}} \exp(\mathbb{E}_{\mathcal{S}^n} \mathrm{I}[\mathrm{sign}(\mathbf{w}^\top\mathbf{c}) = \mathrm{sign}(\mathbf{w}^\top\bar{\mathbf{c}})]).$$

Combining Eq. (29) with Eq. (28) and plugin to Eq. (27), we have

$$4(n-1)^2 \Delta(M_s)^2 = 4(n-1)^2 \left(M_s^{\mathbf{w}}(\phi, \xi) - \widehat{M}_s^{\mathbf{w}}(\phi, \xi)\right)^2$$

$$\leq (2(n-1)(\mathbb{E}_{\mathcal{S}^n} \mathrm{KL}(\hat{P}_s^\phi(\mathbf{C}|\mathbf{X}=\mathbf{x})\|\pi_{\mathbf{C}}) + \mathbb{E}_{\mathcal{S}^n} \mathrm{KL}(\hat{P}_s^\xi(\bar{\mathbf{C}}|\mathbf{X}=\mathbf{x})\|\pi_{\bar{\mathbf{C}}})$$

$$+ \ln \mathbb{E}_{\mathbf{c}\sim\pi_{\mathbf{C}}} \mathbb{E}_{\mathbf{c}\sim\pi_{\bar{\mathbf{C}}}} \exp(2(n-1)\Delta(M_s')))^2,$$

where $\Delta(M_s') = |\mathbb{E}_{\mathcal{S}^n} \mathrm{I}[\mathrm{sign}(\mathbf{w}^\top\mathbf{c}) = \mathrm{sign}(\mathbf{w}^\top\bar{\mathbf{c}})] - \mathbb{E}_{\mathcal{S}} \mathrm{I}[\mathrm{sign}(\mathbf{w}^\top\mathbf{c}) = \mathrm{sign}(\mathbf{w}^\top\bar{\mathbf{c}})]|$. The rest of the proof is similar to Theorem 3.3 (1), thus we get the theoretical results of Theorem 3.3 (2).

# E Satisfaction of Exogeneity

In this section, we provide the proof of Theorem 4.3 and demonstrate why minimizing the objective is equivalent to finding the $\mathbf{C}$ satisfying conditional independence.

The proof process consists of three steps: (i) proof the optimization of

$$\min_{\phi,\mathbf{w}} \widehat{SF}_s(\mathbf{w},\phi) + \lambda \mathbb{E}_{\mathcal{S}^n} \text{KL}(\hat{P}_s^\phi(\mathbf{C}|\mathbf{X}=\mathbf{x})\|\pi_{\mathbf{C}}) \tag{30}$$

is equivalent with the optimization of Information Bottleneck Shamir et al. (2010a) objective $L_{\text{ib}} = \max I(\mathbf{C},Y) - \lambda I(\mathbf{X},\mathbf{C})$, where $I(A,B)$ denotes the mutual information between $A$ and $B$. $\mathbf{X},Y$ are from $\mathcal{S}^n$. (ii) The optimal solution of Information Bottleneck satisfies $\mathbf{X} \perp Y|\mathbf{C}$. To simplify the writing, we represent $P_s(\cdot)$ by $P(\cdot)$.

**For step (i)**: The objective Eq. (30) is coincide with the objective of variational autoencoder (Doersch, 2016) which is proved to be equivalent to the IB objective. The theoretical results are provided in Alemi et al. (2016), and we report the proof process from the notations in our paper. The derivation starts with bound the $I(\mathbf{C},Y)$ and $I(\mathbf{C},\mathbf{X})$ separately, for the term $I(\mathbf{C},Y)$,

$$I(\mathbf{C},Y) = \int P(Y,\mathbf{C}) \log \frac{P(Y,\mathbf{C})}{P(Y)P(\mathbf{C})} dYd\mathbf{C} = \int P(Y,\mathbf{C}) \log \frac{P(Y\mid\mathbf{C})}{P(Y)} dYd\mathbf{C}. \tag{31}$$

Decompose the $P(Y\mid\mathbf{C})$ as below

$$P(Y\mid\mathbf{C}) = \int d\mathbf{X} P(\mathbf{X},Y\mid\mathbf{C}) = \int d\mathbf{X} P(Y\mid\mathbf{X})P(\mathbf{X}\mid\mathbf{C}) = \int d\mathbf{X} \frac{P(Y\mid\mathbf{X})\hat{P}^\phi(\mathbf{C}\mid\mathbf{X})P(\mathbf{X})}{P(\mathbf{C})}.$$

In the process, the $P(Y\mid\mathbf{C})$ is approximated by parameterized $\hat{P}(Y\mid\mathbf{C})$. Since the KL divergence is always larger than $0$,

$$\text{KL}[P(Y\mid\mathbf{C}),\hat{P}(Y\mid\mathbf{C})] \geq 0 \implies \int P(Y\mid\mathbf{C}) \log P(Y\mid\mathbf{C})dY \geq \int P(Y\mid\mathbf{C}) \log \hat{P}(Y\mid\mathbf{C})dY.$$

and hence

$$I(\mathbf{C},Y) \geq \int P(Y,\mathbf{C}) \log \frac{\hat{P}(Y\mid\mathbf{C})}{P(Y)}$$

$$= \int P(Y,\mathbf{C}) \log \hat{P}(Y\mid\mathbf{C}) - \int dY P(Y) \log P(Y)dYd\mathbf{C}$$

$$= \int P(Y,\mathbf{C}) \log \hat{P}(Y\mid\mathbf{C}) + H(Y)dYd\mathbf{C}$$

Then $I(\mathbf{C},Y)$ is lower bounded by follows:

$$I(\mathbf{C},Y) \geq \int P(\mathbf{X})P(Y\mid\mathbf{X})P(\mathbf{C}\mid d\mathbf{X}dYd\mathbf{C}) \log \hat{P}^\phi(Y\mid\mathbf{C})d\mathbf{X}dYd\mathbf{C}$$

Then, consider the term $\lambda I(\mathbf{C},\mathbf{X})$, it is decomposed by following equation:

$$I(\mathbf{C},\mathbf{X}) = \int \hat{P}^\phi(\mathbf{X},\mathbf{C}) \log \frac{\hat{P}^\phi(\mathbf{C}\mid\mathbf{X})}{P(\mathbf{C})} d\mathbf{C}d\mathbf{X}$$

$$= \int \hat{P}^\phi(\mathbf{X},\mathbf{C}) \log \hat{P}^\phi(\mathbf{C}\mid\mathbf{X})d\mathbf{C}d\mathbf{X} - \int P(\mathbf{C}) \log P(\mathbf{C})d\mathbf{C}d\mathbf{X}. \tag{32}$$

It is upper bounded by

$$I(\mathbf{C},\mathbf{X}) \leq \int P(\mathbf{X})\hat{P}^\phi(\mathbf{C}\mid\mathbf{X}) \log \frac{\hat{P}^\phi(\mathbf{C}\mid\mathbf{X})}{P(\mathbf{C})} d\mathbf{X}d\mathbf{C}$$

where $P(\mathbf{C})$ is prior of $\mathbf{C}$. Combining the upper bound of $I(\mathbf{C},\mathbf{X})$ and the lower bound of $I(\mathbf{C},Y)$ together, the lower bound of IB objective is

$$I(\mathbf{C},Y) - \lambda I(\mathbf{C},\mathbf{X}) \geq \int P(\mathbf{X})P(Y\mid\mathbf{X})P(\mathbf{C}\mid\mathbf{x}) \log \hat{P}(Y\mid\mathbf{C})d\mathbf{X}dYd\mathbf{C}$$

$$- \lambda \int P(\mathbf{X})\hat{P}^\phi(\mathbf{C}\mid\mathbf{X}) \log \frac{\hat{P}^\phi(\mathbf{C}\mid\mathbf{X})}{P(\mathbf{C})} d\mathbf{X}d\mathbf{C} \tag{33}$$

From the results, we can draw a conclusion that maximizing the IB objective $I(\mathbf{C}, Y) - \lambda I(\mathbf{C}, \mathbf{X})$ is equivalent to minimizing the objective Eq.(9) in Theorem 4.3.

**For step (ii)**: The proof refers to the theoretical results in Shamir et al. (2010a), they correspond the optimal solution of information bottleneck with minimal sufficient statistics defined below. We first demonstrate the definition of Mininal Sufficient Statistic in Fisher (1922); Shamir et al. (2010a) below.

**Definition E.1** (Sufficient Statistic Fisher (1922); Shamir et al. (2010a)). Let $Y$ be a parameter of probability distributions and $\mathbf{X}$ is random variable drawn from a probability distribution determined by $Y$. of $\mathbf{C}$ is a function of $\mathbf{X}$, then $\mathbf{C}$ is sufficient for $Y$ if

$$\forall \mathbf{x} \in \mathcal{X}, \mathbf{c} \in \mathbb{R}^d, y \in \mathcal{Y} \quad P(\mathbf{X} = \mathbf{x} \mid \mathbf{C} = \mathbf{c}, Y = y) = P(\mathbf{X} = \mathbf{x} \mid \mathbf{C} = \mathbf{c})$$

The definition indicates that the sufficient statistic $\mathbf{C}$ satisfy the conditional independency $\mathbf{X} \perp Y | \mathbf{C}$.

**Definition E.2** (Minimal Sufficient Statistic Fisher (1922); Shamir et al. (2010a)). A sufficient statistic $\mathbf{C}'$ is minimal if and only if for any sufficient statistic $\mathbf{C}$, there exists a deterministic function $f$ such that $\mathbf{C}' = f(\mathbf{C})$ almost everywhere w.r.t $\mathbf{X}$.

From Theorem 7 in Shamir et al. (2010a), the optimization of IB is correspond to finding the minimal sufficient statistic. We show Theorem 7 as a lemme below.

**Lemma E.3** (Shamir et al. (2010a)). *Let $\mathbf{X}$ be a sample drawn according to a distribution determined by the random variable $Y$. The set of solutions to*

$$\min_{\mathbf{C}} I(\mathbf{X}, \mathbf{C}) \text{ s.t. } I(Y, \mathbf{C}) = \max_{\mathbf{C}'} I(Y; \mathbf{C}')$$

*is exactly the set of minimal sufficient statistics for $Y$ based on the sample $\mathbf{X}$.*

From Lemma E.3, the process of optimizing the IB objective $I(\mathbf{C}, Y) - \lambda I(\mathbf{C}, \mathbf{X})$ is equivalent to finding the minimal sufficient statistic. Then, the process of optimizing the IB objective is the process to find a $\mathbf{C}$ satisfying conditional independency $\mathbf{X} \perp Y | \mathbf{C}$.

From Step (i) and (ii), we get the result in Theorem 4.3 that optimizing the objective $\min_{\phi, \mathbf{w}, \lambda} L_{\text{exo}}$ is equivalent to finding a $\mathbf{C}$ satisfying conditional independency $\mathbf{X} \perp Y | \mathbf{C}$. Considering Assumption 4.2 in our paper, we can draw the conclusion that optimizing the objective Eq.(9) is equivalent to satisfying the exogeneity.

# F    Additional Related Works

**OOD generalization.** In addition to the related works in main text, some works (Gong et al., 2016; Li et al., 2018b; Magliacane et al., 2017; Lu et al., 2021; Rojas-Carulla et al., 2018; Meinshausen, 2018; Peters et al., 2016) solve the OOD generalization problem by learning causal representations from source data by introducing invariance-based regularization into cross-domain representation learning and consider the task labels across multi-domains.

**Domain adaptation.** Domain Adaptation aims at learning to generalize to the test domain where the distribution is not the same as source domain, while the data from the test data is available during traning process. Existing works on domain adaptation use a different way to achieve the goal (Pan et al., 2010; You et al., 2019; Zhang et al., 2013). Magliacane et al. (2018) considers solving the generalization problem from the causality perspective and Zhao et al. (2019) propose a framework of invariant representation learning in domain adaptation. Furthermore, Zhang et al. (2019) provides theoretical analysis of domain adaptation.

**Causal discovery.** In order to investigate the relationship between causally related variables in real-world systems, traditional methods often employ principles such as the Markov condition and the principle of conditional independence between cause and mechanism (ICM) to discover causal structures or differentiate causes from effects (Mooij et al., 2016). Additionally, several studies have focused on the inherent asymmetry between cause and effect, as seen in the works of Sontakke et al. (2021); Steudel et al. (2010); Janzing & Schölkopf (2010), and Cover (1999). These ideas have also been utilized by Parascandolo et al. (2018) and Steudel et al. (2010). Other research lines focus on modeling structural causal models and discussing the identifiability of causal structures and functions (Hoyer et al., 2008; Zhang & Hyvarinen, 2012). The task of causal discovery typically assumes that

all variables are observable, aiming to identify the unknown causal graph. In contrast, our work deals with causal factors that are hidden within observational data, given a known causal graph. Our objective is to extract causal information from the observational data.

**Causal Disentangled Representation Learning**: Causal representation learning methods, such as those discussed in Schölkopf et al. (2021), aim to find the disentanglement representation from observational data with causal semantic meaning. Previous works have utilized structural causal models to capture causal relationships within complex observational data, as seen in the works of Yang et al. (2021); Shen et al. (2020); Lu et al. (2021); Wang et al. (2020); Yang et al. (2023), which focus on disentangling causal concepts from the original inputs. While our work also aims to specify causal representations from observational data, we distinguish ourselves by focusing specifically on identifying sufficient and necessary causes. Wang & Jordan (2021) concentrate on learning to disentangle causal representations using non-spurious and efficient information from observational data. Their framework is inspired by the concept of PNS and extends it to a continuous representation space. In contrast, our learning framework is based on the task of OOD generalization, and our algorithms consider both monotonicity and exogeneity conditions. Additionally, we provide a theoretical analysis of the generalization ability of our approach based on PAC learning frameworks, as explored in the works of Shalev-Shwartz & Ben-David (2014) and Shamir et al. (2010b).

**Contrastive Learning**: The proposed method is also related to another line of research called contrastive learning. These works often deal with the OOD generalization task by designing an objective function that contrasts certain properties of the representation. Saunshi et al. (2022) argue that the framework learns invariant information by introducing certain biases, and the core problem of contrastive learning is the objective function (Xiao et al., 2020). Different from their approaches that perform learning by contrastivity, we use counterfactual PNS reasoning as the function to be optimized, which learns invariant causal information with an explicit stability guarantee for the model's performance.

# G   Broader Impacts

In this paper, we propose a method to specify the more critical representations based on invariant learning, where we extract the representations from the invariant information that possess sufficient and necessary causalities for the prediction task. This effectively helps improve the performance and robustness of machine learning models in unknown domains. The application of this method can promote reliability and usability in various real-world scenarios. Furthermore, through experiments on real-world datasets, our method demonstrates the ability to enhance model performance even in the worst-case scenarios. This highlights the potential of our method to enhance reliability and usability in various out-of-distribution (OOD) settings. By extracting the essential causal representations from the invariant information, our method provides valuable insights into the underlying causal relationships in the data, leading to improved generalization and robustness in OOD scenarios. These findings support the broader impact of our approach in enhancing the reliability and usability of machine learning models across different domains and applications.

