# OpenReview forum: "Invariant Learning via Probability of Sufficient and Necessary Causes"
_NeurIPS.cc/2023/Conference — NeurIPS 2023 spotlight_

### Official Review · Reviewer_CQCE · 2023-07-01

**Soundness:** 3 good
**Presentation:** 3 good
**Contribution:** 3 good
**Rating:** 7
**Confidence:** 1

**Summary:**

This paper proposes the new concept of 'essential causal information' to achieve OOD generalization and proposes the probability of necessarity and sufficiency (PNS) risk to estimate the representation if satisfies the new concept. Then, it proposes a new algorithm to achieve loss PNS risk on the test domain.

**Strengths:**

(1) Introducing the PNS risk for out-of-distribution generalization is novel and interesting. The proof for it is rigorous.
(2) The method is general and works well on various datasets.
(3) The writing of this paper is good.

**Weaknesses:**

(1) The fox also has long whiskers. The example in section 2.3 does not hold on.

**Questions:**

Why you don't consider recent methods (2021~2023) for Domain Adaptation? I am not an expert in this domain but I think there have many strong and new baselines for out-of-distribution generalization.

**Limitations:**

I don't see any potential negative social impact of their work.

---

> ### Author Rebuttal · Authors · 2023-08-09
>
> ## Response to Reviewer CQCE:
>
> We appreciate the suggestions from reviewer CQCE and are grateful for the recognition of the novelty of our approach, the rigor of our theory, and our writing style. We will respond to each of the questions posed by the reviewer below.
>
> **Q1 The fox also has long whiskers. The example in section 2.3 does not hold on.**
>
> Thank you for your insightful question. We presented this example in the paper to provide a more vivid understanding of PNS. To strengthen our example, in revision, we will modify the example of 'sufficient and necessary cause' in the context of predicting cats and foxes by the feature of 'long mouth' instead of 'whisker'.
>
> **Q2 Why you don't consider recent methods (2021~2023).**
>
> Thank you for your constructive suggestion.
> Our main method primarily focuses on Domain Generalization. Additionally, we included a recently proposed causal-based method EQRM[1] proposed in 2022, and its results on  VLCS datasets are shown below.
>
> |     | C  |  L    | S  | V  |Avg |Min   |
> |  ----  | ----  |----  |----  |----  |----  |----  |
> | ERM  | 97.7±0.4| 64.3±0.9 | 73.4±0.5 |74.6±1.3|77.5|64.3
> | IRM  | **98.6±0.1**|64.9±0.9|**73.4±0.6**|**77.3±0.9**| 78.5 |64.9|
> | EQRM  | 98.3±0.0 | 63.7±0.8 | 72.6±1.0 | 76.7±1.1| 77.8| 63.7|
> | CaSN(ours)  |98.3 ±0.3 |**66.9±1.2** | 73.1±0.6 | 76.8±0.8 |**78.8** | **66.9**|
>
> The results show that our method has a competitive performance compared with the recent methods on EQRM. We plan to implement more recent baseline methods and test them on more datasets. The results will update in revision.
>
> Our extended algorithm for Domain Adaptation is not the main focus of our paper. Instead, we provided experimental validation of its effectiveness in comparison with classical algorithms in the appendix. We plan to implement the recent Domain Adaptation method based on the DomainBed code. We plan to update the relevant results in the revision.
>
>
> In addition，we test our method on the recent dataset SpuCo[2] in 2023 and compare our method CaSN with the benchmark they provided. We obtained some new results and the code implementation is based on [SpuCo](https://github.com/BigML-CS-UCLA/SpuCo). Our experimental data is SpuCoAnimal which comprises 8 different spurious correlation groups.
>
> We set hyperparameters as $\delta=3$ and $\lambda\in{0.005,0.01}$, while the remaining parameters followed SpuCo's default values. We conducted two types of experiments: the first involved training the model without providing any additional information, and the second involved resampling based on the spurious correlation group information, where the base models corresponded to ERM and Group Balance (GB) respectively.
>
> We reported the results below, with each result obtained from 4 fixed random seeds.
>
>
> |     | ERM  |  CaSN (ours)   | GB  | CaSN-GB  |
> |  ----  | ----  |----  |----  |---- |
> |  Average  | 80.26±4.1  |84.34±0.6   |49.9±1.4 | 50.54±7.6 |
> |  Min  | 8.3±1.3  | 9.7±2.7   |41.1±2.7| 42.0±0.8 |
>
> Our method showed higher average accuracy compared to the baseline, and even the worst spurious correlation group exhibited higher accuracy than the baseline.  We plan to continually update the results in revision.
>
> [1] Eastwood C, Robey A, Singh S, et al. Probable domain generalization via quantile risk minimization[J]. Advances in Neural Information Processing Systems, 2022, 35: 17340-17358.
>
>
> [2] Joshi S, Yang Y, Xue Y, et al. Towards Mitigating Spurious Correlations in the Wild: A Benchmark & a more Realistic Dataset[J]. arXiv preprint arXiv:2306.11957, 2023.

---

### Official Review · Reviewer_pPMP · 2023-07-04

**Soundness:** 3 good
**Presentation:** 3 good
**Contribution:** 3 good
**Rating:** 7
**Confidence:** 4

**Summary:**

This paper presents a novel approach to enhance out-of-distribution (OOD) generalization by identifying both sufficient and necessary conditions, thereby extending the existing literature that primarily focuses on causal invariance. The proposed method employs the probability of sufficient and necessary causes (PNS) to demonstrate its effectiveness in improving OOD generalization, through theoretical analysis and experimental results,.

**Strengths:**

1. The proposed method is very intuitive and novel, as far as I know. It focuses on sufficient and necessary causes to explore the invariance. This method successfully combines existing ideas in invariant causality with PNS, and has good experimental results.

2. The proposed delta-PNS is reasonable to me. If the semantics have overlap, it is impossible to make good prediction.

3. The experiment part is extensive which covers a variety of common datasets, including synthetic data, an ablation study, and datasets for both OOD generalization and domain adaptation tasks.

**Weaknesses:**

1. It is difficult to understand the definition of PNS in Definition 2.1, Could you provide more details?

2. Typo: missing ) in line 222.

3. Do you have discussed more general causal structure?


-------

After reading the author's response, my concerns have been well addressed, so I will keep my score as accept.

**Questions:**

Please answer the questions in Weakness.

---

> ### Author Rebuttal · Authors · 2023-08-09
>
> ## Response to Reviewer pPMP:
> We thank reviewer pPMP for the constructive feedback and recognition of our novel and intuitive idea, as well as our experiments on various datasets. We respond to each of the reviewer's questions in the sections below.
>
> **Q1 It is difficult to understand the definition of PNS in Definition 2.1, Could you provide more details?**
>
> According to your valuable question, we will highlight the following details in the revision to enhance understanding of PNS.
> Considering that the definition of PNS is based on the counterfactual distribution, it is not inherently intuitive. To better grasp how PNS is calculated and its relation to sufficiency and necessity, we provide two examples in General Response's G.P.1.
>
> From the example, we demonstrate how to calculate the PNS value of a feature in a predictive task and establish how a sufficient but unnecessary cause aligns with a specific PNS value. This straightforward case suggests that the PNS value can distinguish the extent to which a feature is necessary or sufficient.
>
> To enhance understanding of PNS, we will provide more descriptions in the revision.
>
>
> **Q2 Typo: missing ) in line 222.**
>
> Thank you for your suggestion; we will diligently review and correct typos to enhance the quality of the paper.
>
>
> **Q3 Do you have discussed a more general causal structure?**
>
> Thank you for your question. The various assumptions in causality have always been a topic worth discussing. Our paper's assumption is based on conditional independence $X\perp Y|C$, following the PIIF in [1], which is widely accepted. In [1], they also discuss other assumptions, such as FIIF, where the conditional independence $X \perp Y \mid C$ no longer holds, but it has assumption $Y \perp \mathcal{E} \mid C$, where $\mathcal{E}$ denotes the variable of domain). The IRM-based approach [2] effectively addresses generalization issues under this assumption but fails on PIIF assumption $X\perp Y|C$.
>
> In Section 7, we briefly discuss potential extensions of our paper to accommodate a broader range of causal assumptions, such as PIIF, confounders, and anti-causal. For instance, under the PIIF assumption, conditional independence should be described as $Y \perp \mathcal{E} \mid C$. To meet Exogeneity under this assumption, an objective function (e.g., the invariant constraint in IRM) that incorporates additional domain label information is required. Further, when considering the existence of a confounder in causal assumption, an auxiliary objective function must be designed to identify the confounder, ensuring the satisfaction of Exogeneity.
>
> More details about causal assumptions are concluded in General Response's G.P.2. We will provide more descriptions in the revision.
>
> [1] Ahuja et al. Invariance principle meets information bottleneck for out-of-distribution generalization.
>
> [2] Arjovsky et al. Invariant risk minimization.

---

> > ### Comment · Reviewer_pPMP · 2023-08-21
> >
> > Thank you for the detailed response. My concerns have been well addressed, so I will keep my score as accept.

---

> > > ### Author Response · Authors · 2023-08-21
> > >
> > > We sincerely appreciate your efforts in reviewing our rebuttal and offering feedback. We will continuously improve the quality of the paper. Thank you!

---

### Official Review · Reviewer_qVZ2 · 2023-07-05

**Soundness:** 2 fair
**Presentation:** 3 good
**Contribution:** 3 good
**Rating:** 5
**Confidence:** 3

**Summary:**

The paper proposes a new formulation of the out-of-distribution generalization task via finding necessary and sufficient features, similar to invariant feature learning but with some added constraints.
The paper uses the definition of necessary and sufficient from Pearl (2008) and proposes a related risk measure.
Then, a connection between source and test risks is given along with assumptions needed for the bounds to hold.
Finally, the paper proposes a practical adversarial algorithm to minimize the proposed objective and presents empirical results on DomainBed and a toy simulated dataset.


**Strengths:**

- The paper proposes an interesting new direction to find necessary and sufficient features and proposes a risk based on this objective.

- The paper theoretically connects the target risk and source risk based on the theoretic objective.

- The paper provides reasonable results on PACS and VLCS and a toy simulated dataset to validate the theory.


**Weaknesses:**

- PNS needs to be explained and discussed more. This is a very difficult concept to grasp theoretically. Also, Definition 2.1 seems incomplete. What are $c$ and $\bar{c}$? How can just two realizations be used to determine necessity and sufficiency? Is it some expectation over $c$ and $\bar{c}$? Even theoretically, how would I come up with $c$ and $\bar{c}$?  After looking at (Pearl 2009), it seems that it only deals with binary variables which would make more sense. Is this an extension of (Pearl 2009)? If so, this needs to explained more.

- What is the distribution of the $\bar{C}$ variable (even in theory)? This again is important missing detail. This is not explained except that there is another auxiliary distribution. It seems to be a completely independent random variable but it is unclear what distribution it is or should be in theory (or in practice).

- It is not explained why the monotonicity term can be "absorbed" into the monotonicity measurement (Line 144). Also, why does this make sense as a monotonicity term? Why would this encourage monotonicity? Again, part of the confusion is that it is completely unclear what the distribution of $\bar{C}$ should be in theory.

- For equation 6, is $\mathcal{T}(x,y) = P_t(x,y)$? If so, then this is the expected ratio to the $k$-th power? But this isn't a divergence because if $\mathcal{T} = \mathcal{S}$ this "divergence" would be 1. Thus, it does not even satisfy the most basic property of a distribution divergence. I also could not find the definition of $\beta$ divergence quickly in (Gong et al. 2016). Could you reference the specific definition?

- Again, assumption 4.1 is unclear. And perhaps this could aid in answering the questions above. But it seems that the distributions of $C$ and $\bar{C}$ need to be different. Yet, again, I am not sure why as the paper does not explain $\bar{C}$ well.

- Theorem 4.3 contains typos and seems to be incomplete. The statement after the equation is not a complete sentence and does not match the previous sentence.

- The paper lacks a direct comparison with related works like IRM. How is the proposed objective or algorithm fundamentally different or similar to IRM or related methods? A direct comparison of the final equations would aid in understanding how this work fits in with related work.

- The paper lacks discussion on the limitations of the practical optimization algorithm especially since this is a new adversarial optimization algorithm. See limitations comments.

**Minor:**
- The examples of necessary and sufficient features could be clarified. Perhaps just a 2 x 2 table of with necessary/ not necessary and  sufficient/not sufficient. And then, just fill in each box with a feature. This might be simpler than having an example that takes careful analysis. Also perhaps showing with arrows, it might be easier to show. Like whiskers <-> cat, cat feet -> cat, pointy ears <- cat...

- The figure is in the completely wrong place. Figures should always be placed after they are referenced and not before.

**Review summary**
Overall, I think the paper could be very good. The perspective is fresh and introduces some new concepts. However, the lack of conceptual and notational clarity and other weaknesses hinder it from being a strong paper.


**Questions:**

- Do you limit the number of hyperparameter choices to a fixed number for every method? I believe DomainBed (or maybe WILDS) fixes the number of hyperparameter choices allowed for each method. This means that methods with more hyperparameters are harder to fit and thus may perform worse on average.

- What is PA in Figure 2? I'm guessing that is a typo and should be SN?

- Why not call OOD Generalization the more standard term of "domain generalization"?


**Limitations:**

- The paper does not discuss the practical limitations or computational complexity of the method. The paper does not acknowledge that adversarial learning is inherently a very difficult problem both theoretically and practically. Were any instabilities encountered during training? How were these handled? It seems there were two encoders trained but this would seem to increase training instability. What about the constraint in Eq 8? How is that enforced in practice? Again, this seems to be a very difficult constraint to maintain. Furthermore, the KL term likely requires the distributions to be Gaussians to allow closed for computation, or did you use something else?

- The paper acknowledges some of the limitations with respect to theory. More discussion on the assumptions and which are reasonably likely to hold in practice or not would be helpful. Also, how bad will the theory break if the assumptions do not hold?

---

> ### Author Rebuttal · Authors · 2023-08-09
>
> # Response to Reviewer qVZ2:
>
> We appreciate reviewer qVZ2's constructive feedback and positive comments on our idea, theoretical contributions, and experiments contributions. We respond to each of the reviewer's questions in the following.
>
> **Q1.1**  **Need to discussed more to understand PNS**
>
> We conclude the answer in G.P.1 of General Response.
>
> **Q1.2**  **What are $c$ and $\bar{c}$? if Def 2.1 is an extension of (Pearl 2009)?**
>
> In the definition of PNS (Definition 2.1), $c$ and $\bar{c}$ are different samples of variable C. In our study, we extend the concept of PNS to cases where C is not binary. In such scenarios, PNS on c and $\bar{c}$ can still describe sufficient and necessary. We conclude a more detailed discussion in Example.2 in General Response G.P.1
>
> **Q2&Q5**  **It is not clear about the $\bar{C}$ in theory and Assumption 4.1.**
>
> Thank you for pointing out the unclear definitions in the paper. We detailedly explain it here.
> $\bar{C}$ is an auxiliary variable to describe the intervention samples obtained by sampling from a specific distribution (lines 129-133). 'Intervention' describes we force a variable to be a certain value (may be different from its natural value), it is commonly accepted that the distribution of intervention data can be different from the original distribution. In our paper, the distribution of $\bar{C}$ is set to be different from the distribution of $C$. Generally, they are different distributions, but in the same distribution class, e.g., different distributions from the Gaussian distribution class.
>
>
> We can understand Assumption 4.1 as stating that there should be some distance between $c$ and $\bar{c}$ if they lead to different semantics on Y. The constraint $|\bar{c}-c|>δ$ in Eq. 8 actually ensures that if Assumption 4.1 holds, original value $c$ and intervention $\bar{c}$ should have a distance to make c predicts y and avoids $\bar{c}$ predicting y.
>
>
> **Q3**  **Why NC term can be "absorbed" into the monotonicity measurement?**
>
> In Appendix D1, we provide proof that the Monotonicity Measurement ($M_t$) can be decomposed into a combination of two components (SF and NC). The key proofs are in Eqs. 15&16. It is evident that the optimization of the NC component is implicitly included in the decomposition of $M_t$.
>
>
> **Q4**  **Is $T(x, y)=P_t(x, y)$? What is the detailed property of $\beta$-divergence?**
>
>
> Yes, $T(x, y)=P_t(x, y)$. The definition of $\beta_k$ is from Eq. 7 in Germain et. al.[1]. $β_k$ is a transformation of widely used Renyi divergence (footnote on page 4 of [1]), the relationship is,
>
> $for~ k\ge 0, \log β_k(T|S)=\frac{k-1}{k}{{D_k}(T|S)}.$
>
> $\log β_k$ will take the lowest value 0 when $T=S$. There was a reference typo in the main text that mistakenly cited Gong et al. We'll address this and correct the typo in the revision as Germain et. al.[1].
>
>
>
> **Q6**  **Theorem 4.3 contains typos and seems to be incomplete**
>
> We will modify Theorem 4.3 as below to make it more fluent.
>
> Theorem 4.3. The optimal solution of C was learned by the following objective
> ...$X ⊥ Y | C$.
>
>
> **Q7**  **The paper lacks a direct comparison with related works like IRM.**
>
> We conclude the answer in G.P.2 of General Response.
>
> **Q9**  **Do you limit the number of hyperparameter choices to a fixed number for every method?**
>
> Yes, during the training process, we limit the number of hyperparameters choices. We list the range of hyperparameters in Appendix Table 3. We utilized DomainBed's sweep script to run experiments and set the number of hyperparameter choices (--n_hparams) as 10.
>
>
>
> **Questions.** **Typos and figure re-ognization**
>
>
> 1. We will follow your suggestion and modify the presentation of examples in  Figure 1 to make them clearer.
> 2. We will correct the typo in Figure 2 and replace "PA" with "SN."
> 3. As per your advice, we will consider changing the phrasing from "OOD generalization" to "domain generalization" to avoid any ambiguity.
>
> **L.1**  **Should discuss the practical limitations of adversarial objective and report computational complexity of the method**
>
> Thank you for your suggestion! We compared the time complexity of our method with other approaches. On the Colored-MNIST dataset, the average runtime per 100 steps across 3 domains is CaSN(ours): 19.5ms, ERM: 5.9ms, IRM: 8.1ms, and MLDG: 17.8ms. The differences in runtime mainly stem from the separate optimization steps for the min and max optimization processes. While our method's runtime is higher than ERM and IRM, it is still acceptable compared to classical domain generalization methods like MLDG.
>
> Regarding adversarial learning, we lower the learning rate and fewer steps for maximization to make the training process stable. Inspired by your valuable question, we surprisingly find that introducing a warm-up scheme can further stabilize the training process. Namely, we initially exclude the max step and pre-train all the components including parameters of $\bar{C}$ in the minimization step. We will highlight these experimental details in our revision.
>
> For the constraint in Eq. 8, we optimize it by a Lagrange multiplier. Specifically, during training, we added an additional objective in both the min and max steps as $λ min||\bar{c}-c|_2-δ|_2$. This constraint is not a hard constraint. In our experiments, this constraint makes $|\bar{c}-c|_2$ approach $\delta$. We set δ as a hyperparameter, and the experimental results in Figure 2 showed that smaller values of δ are less conducive to discovering SN information, which aligns with our Assumption 4.1.
>
> We will include more discussion on the implementation details and limitations of the practical algorithm in the revision.
>
>
>
> **L.2**  **More discussion on the assumptions and which are reasonably likely to hold in practice or not would be helpful**
>
> We conclude this answer in G.P.2 of General Response.
>
>
> [1] Germain et. al. A new pac-bayesian perspective on domain adaptation.

---

> > ### Comment · Reviewer_qVZ2 · 2023-08-12
> >
> > Thank you for your response. I appreciated the added examples and further clarity. I would encourage the authors to push more for this in the final version if accepted. These are new concepts that need to be explained well and can be confusing. Both precise and clear should be the goals. Given both the general response and specific response, I have raised my score.

---

> > > ### Author Response · Authors · 2023-08-21
> > >
> > > Thank you for taking the time to consider our rebuttal and providing further feedback! We deeply appreciate your constructive comments and we will continue to make improvements based on your suggestions!

---

### Official Review · Reviewer_jeo8 · 2023-07-06

**Soundness:** 4 excellent
**Presentation:** 3 good
**Contribution:** 4 excellent
**Rating:** 8
**Confidence:** 4

**Summary:**

A mostly theoretical paper that formulates the necessity and sufficiency of causal variables in order to optimize models to rely on necessary & sufficient variables. The paper provides rigorous theory connecting the empirical loss when models are optimized using the Probability of Necessity and Sufficiency Loss (PNS). Finally, there are some empirical results verifying the effectiveness of this approach.

**Strengths:**

1) Rigorous theoretical guarantees for OOD generalization when optimizing with the introduced PNS loss on source domain S. Builds upon prior well-established theory.

2) Clear and well-explained intuition of using necessary and sufficient causal variables to prevent performance degradation when generalizing to unseen domains


**Weaknesses:**

1) Rigorous empirical evaluation in the presence of more realistic datasets could help this paper be even more convincing. Evaluating on the recent more challenging dataset SpuCo for spurious correlations where existing methods fail can be a good starting point. Alternatively, considering other datasets from the WILDS package would be another useful direction.

**Questions:**

N/A

---

> ### Author Rebuttal · Authors · 2023-08-09
>
> ## Response to Reviewer jeo8:
>
> We thank reviewer jeo8 for their suggestions and appreciate their recognition that our method is intuitive and our theory is rigorous. We will address each of the questions posed by the reviewer in the sections below.
>
> **Questions. Rigorous empirical evaluation in the presence of more realistic datasets like SpuCo and WILDS could help this paper be even more convincing.**
>
> Thank you for your suggestion! We obtained some new results on SpuCo[1] based on the code [SpuCo](https://github.com/BigML-CS-UCLA/SpuCo). Our experimental data is SpuCoAnimal which comprises 8 different spurious correlation groups.
>
> We set hyperparameters as $\delta=3$ and $\lambda\in{0.005,0.01}$, while the remaining parameters followed SpuCo's default values. We conducted two types of experiments: the first involved training the model without providing any additional information, and the second involved resampling based on the spurious correlation group information, where the base models corresponded to ERM and Group Balance (GB) separately. We also include GroupDRO[2] as one of the baselines, the baselines performance is reported SpuCo[1]
>
> Data augmentation-based methods, ERM* and GB*, were not included in these experiments.
>
> We reported the results below, with each result obtained from 4 fixed random seeds.
>
> |     | ERM  |  CaSN    | GroupDRO| GB  | CaSN-GB  |
> |  ----  | ----  |----  |----  |---- | ---- |
> |  Average  | 80.26±4.1  |84.34±0.6  | 50.7±4.2 |49.9±1.4 | 50.54±7.6 |
> |  Min  | 8.3±1.3  | 9.7±2.7  | 36.7±0.6 |41.1±2.7| 42.0±0.8 |
>
> The baseline results are directly taken from SpuCo[1].  Our method showed higher average accuracy on average accuracy compared to the baseline, and even the worst spurious correlation group exhibited higher accuracy than the baseline. We plan to continually update the results by sufficient hyperparameter tuning in revision.
>
> Regarding the WILDS dataset, it is too large and we have not been able to obtain results due to computational resource limitations. We will report the WILDS dataset results in the revision.
>
> [1] Joshi S, Yang Y, Xue Y, et al. Towards Mitigating Spurious Correlations in the Wild: A Benchmark & a more Realistic Dataset[J]. arXiv preprint arXiv:2306.11957, 2023.
>
> [2] Sagawa S, Koh P W, Hashimoto T B, et al. Distributionally robust neural networks for group shifts: On the importance of regularization for worst-case generalization[J]. arXiv preprint arXiv:1911.08731, 2019.

---

> > ### Comment · Reviewer_jeo8 · 2023-08-10
> > **Authors have addressed concerns (to some degree) regarding lack of larger-scale experiments**
> >
> > These are promising empirical results that I think further strengthen the paper and should definitely be included in the revision.
> >
> > I stand by my original assessment of the paper and recommend for acceptance.

---

> > > ### Author Response · Authors · 2023-08-21
> > >
> > > We are genuinely grateful for the diligence you've shown in assessing our rebuttal and providing further comments. We will continuously improve the quality of the paper based on your constructive suggestions. Thank you!

---

### Official Review · Reviewer_aG69 · 2023-07-28

**Soundness:** 3 good
**Presentation:** 2 fair
**Contribution:** 3 good
**Rating:** 7
**Confidence:** 4

**Summary:**

This paper addresses the crucial issue of out-of-distribution (OOD) generalization in machine learning models, where the testing data distribution differs from the training data. To tackle this limitation, the authors introduce the concept of the Probability of Sufficiency and Necessary Causes (PNS), representing the likelihood of a feature being both necessary and sufficient. They propose a novel optimization approach, PNS risk, to enhance representations with high PNS values, effectively capturing the information of sufficient and necessary causes. The paper provides theoretical analysis and experimental results on synthetic and real-world datasets.

**Strengths:**

* The authors propose a novel PNS risk and build the theoretical connections between PNS risk and generalization errors under some assumptions.
* An approximate algorithm is derived to optimize the proposed risk.
* Experimental results validate the effectiveness of PNS risk.

**Weaknesses:**

I have the following comments:
* The difference between the proposed PNS risk objective and other invariant learning methods: Although the PNS risk objective is newly incorporated in invariant learning, I would like to know the key difference from existing metrics/objectives. Since the objective is also to minimize the upper bound of PNS. I am confused about this.
* The exact meaning of the derived bounds: In Equation (7), the authors provide an upper bound of the generalization error on the target distribution. I have some concerns about the RHS: when $k\rightarrow \infty$, $\beta_k$ will be extremely large, and it seems that the proposed method could not address the covariate shift problem. Therefore, I double the real effect of this bound.
* Approximation of algorithms: In algorithms, conditional distributions are parameterized by some parameters, could you analyze the approximation error of such parameterizations? It seems that this is a very theoretical work, but there is a gap between theories and algorithms. Also, the authors do not mention how to calculate $\mathbb E_{c\sim P(C|X)}$, does this require multiple implementations $C$ for one given input $X$? Also, in Equation (8), why the minimization is taken over $\phi, w$?
* Mismatch between algorithmic design and experiments: I think this algorithm & theories are not designed for covariate shifts, which could be seen from the remaining $\beta_k) term in Theorem 3.2. But for image classification tasks, the shift patterns are mainly covariate shifts, since one image is not likely to be given different labels in different domains. Also, for image datasets, the supports of different domains are likely to be disjoint, then why the proposed method could achieve good performance in these settings, which are not well-supported by theoretical analysis.

**Questions:**

Please refer to weaknesses.

**Limitations:**

Please refer to weaknesses.

---

> ### Author Rebuttal · Authors · 2023-08-09
>
> # Response to Reviewer aG69:
> We would like to thank reviewer aG69's comments for our novelty as well as our experiments to validate the effectiveness of the proposed PNS risk. We respond to each of the reviewer's questions in the following.
>
> **Q1.**  **The differences between existing PNS risk and other invariant learning methods.**
>
> We conclude the answer in G.P. 2 of General Response.
>
> **Q2.1**  **When $k →
>  ∞, β_k$ will be large.**
>
> The definition of $β_k$ is from Eq. 7 in [1], namely, $β_k$ is a transformation of widely used  **Renyi divergence** $D_k(T|S)$ (footnote in page 4 of [1]):
>
> $β_∞(T|S)=sup _{(x, y) \in SUPP(S)}(\frac{T(x, y)}{S(x, y)})$
>
>  We can see that when $k →
>  ∞, β_∞(T|S)=2^{{D_∞(T|S)}}$. When the source domain support includes the target one, the divergence does not tend to infinity as $k → ∞$. This proposition is similar to KL divergence.
>
> **Q2.2**  **The method based on Theorem 3.2 could not address the covariate shift problem**
>
> Inspired by your valuable question, we will highlight the following explanations in our revision. Given that in the domain generalization problem, the target domain is unavailable, $β_k$ and $η$ cannot be optimized directly. Thus, instead of directly optimizing the term $β_k$, we address the domain generalization problem based on theoretical results in Theorem 4.3, which shows that the algorithm can identify the invariant features. Specifically, the logic is as follows:
>
> - i) In Section 4.3, we describe that our paper is based on the assumption of conditional independence $X⊥Y∣C$ (PIIF assumption in[4]), where $C$ is invariant variable.
> - ii) Theorem 4.3 describes that the proposed algorithm makes representation satisfy such conditional independence. The proof of Theorem 4.3 indicates the learned representation is minimal sufficient statistics of $Y$ (Definition E.2 in Appendix). Therefore, the learned representation identifies the essential information relative to $Y$ in cause $C$ theoretically.
>
> Combining the above two points, the conclusion of Theorem 4.3 implies that the invariant variable can be learned by objective function Eq. 8.
>
> In discussions of Theorem 3.2, including Eq. 7, we express that invariant feature causes $β_k$ to decrease and narrows the bound. Theorem 4.3 suggests that it's theoretically possible to learn an invariant representation through the objective function. In domain generalization, even though the bounds may account for the divergence between training and target domains, the prevalent approach is to learn generalization by learning invariant representations rather than directly optimizing such divergence, as mentioned in Section 3.2.2 [2].
>
> **Q3.1**  **The approximation error of parameters P(C|X).**
>
> From a causal perspective, given that the ground truth for C is not provided during training, our primary concern is the identifiability of C rather than the approximation error. Theorem 4.3 is key to identifiability. Satisfying identifiability ensures that, given infinite model capacity, there won't be a significant gap between approximation error and true error.
>
> In experiments, our synthetic data provides ground truth information of C. Results in Figure 2 present that we can approximate SN (PA is a typo), SF and NC (information of C).
>
> **Q3.2 How to calculate $E_c$?**
>
> Thank you for your question. We did not elaborate $E_c$ in our paper detailedly, which may lead to misunderstandings. The expectation is on the distribution of C parameterization by a certain $ϕ$. The mathematical formulation of inferenced representation distribution and its expectation is commonly used and aligns with the general approach of Variational Inference, such as the Variational Autoencoder(VAE).
>
> Our implementation mirrors VAE: we first infer the Gaussian distribution's parameter of representations through an encoder (lines 654-656 of appendix), then sample from this distribution. The expectation is calculated on c but not on $ϕ$. We will add discussion to prevent confusion in revision.
>
> **Q3.3**  **Why the minimization is taken over $ϕ, w$?**
>
> $ϕ，w$ and $ξ$ denote the parameters of the representation encoder, classifier, and intervening distribution. Minimizing $ϕ，w$ aims to infer a better representation that can correctly predict label.
>
> **Q4.1**  **Mismatch between algorithm and experiments. This algorithm & theories are not designed for covariate shifts. Why good performance?**
>
> Thank you for pointing out this potentially confusing problem. We will give detailed explanations for the priority of our proposed algorithm, i.e., our algorithm is designed for the mentioned shift.
> - Algorithm design. The causal assumption $X⊥Y|C$ indicates that given invariant feature $C$, $X$ and $Y$ are independent. Specifically, the distribution shift caused by domain-variant features is unrelated to $Y$. Theorem 4.3 suggests our algorithm theoretically satisfies $X⊥Y|C$ and learns the invariant representation.
> - Algorithm eveluation. We experimentally chose covariate shifts to validate our method since, within such shifts, Y remains unchanged despite domain variations if a certain image is provided. A good performance on covariate shifts implies our algorithm can empirically identify representations satisfying $X⊥Y|C$.
>
> **Q4.2** **The supports of different domains are likely to be disjointed.**
>
> In our paper, we considered the domain disjoint problem following[1]. We introduce the term $η_{t /\ s}$. If the support sets of two domains match, this term becomes zero. Similar to $\beta_k$, the term $η_{t /\ s}$ cannot be optimized directly in domain generalization, but it will lower if there are invariant features.
>
> [1] Germain et.al. A new pac-bayesian perspective on domain adaptation.
>
> [2] Wang et.al. Generalizing to Unseen Domains: A Survey on Domain Generalization.
>
> [3] Shen et al. Towards out-of-distribution generalization: A survey.
>
> [4] Ahuja et al. Invariance principle meets information bottleneck for out-of-distribution generalization.

---

> ### Author Response · Authors · 2023-08-17
> **Rebuttal Feedback**
>
> Dear Reviewer aG69
>
> As the discussion period ends soon, we just wanted to check if the response clarified your questions. Thanks again for your constructive feedback.
>
> Best,
> Authors

---

> > ### Comment · Reviewer_aG69 · 2023-08-17
> >
> > The authors have well addressed my concerns, and I hope these discussions could be included in the final version. I have updated my score to 7. Sorry for the delay!

---

> > > ### Author Response · Authors · 2023-08-21
> > >
> > > Thank you for taking the time to review our rebuttal and adjusting the score. We genuinely appreciate your constructive feedback and will continue to make improvements based on your suggestions!

---

### Author Rebuttal · Authors · 2023-08-09

# General Response:
We sincerely appreciate all reviewers's great efforts on review and comments of our work. We thank for the positive comments:
- **Clear and well-explained  intuition** (Reviewer #jeo8 #pPMP )
- **Interesting and reasonable idea** (Reviewer #qVZ2 #pPMP #CQCE)
- **Novel idea** (Reviewer #CQCE #aG69)
- **Rigorous theoretical guarantees** (Reviewer #jeo8 & #qVZ2 #CQCE)
- **Reasonable results** (Reviewer #qVZ2 #pPMP #CQCE #aG69)
- **Good writing** (Reviewer #CQCE)
We provide detailed responses in response to your valuable comments. Due to space limitations, we have listed the common responses first.

**G.P. 1. Understanding PNS**
We provided two examples to help understand the PNS (Probability of Necessity and Sufficiency) value. We will add the following explanations to our revision.

**Example.1** We use the feature 'has cat legs' represented by the variable C (taking values 1 or 0) to predict the label of being a cat or a fox. 'has cat legs' is sufficient but unnecessary cause because the image contains cat legs must have cat but cat image might not contains cat leg. We assume P(Y=1|do(C=1))=1 and P(Y=0|do(C=0))=0.5.

Now, applying the concept of the probability of sufficiency and necessity, we obtain:

Probability of necessity: P(Y(do(C=0)=0|Y=1,X=1)=0.5

Probability of sufficiency: P(Y(do(C=1)=1|Y=0,X=0)=1

In this example, we can state that variable C has a probability of being a sufficient cause.

**Example.2** If we use feature 'eye size' C (taking values 1, 0.5 or 0, the lower value means smaller eye size on face), to predict Y (cat 1 or fox 0). Assuming P(Y=1|do(C=1))=1, P(Y=1|do(C=0.5))=0.5 and  P(Y=1|do(C=0))=0.

In Definition 2.1, PNS($c, \bar{c}$) can take multiple choice. We provide Case.1: $c=1$ and $\bar{c}=0.5$ and Case.2: $c=1$ and $\bar{c}=0$,

Case.1 PNS(1, 0.5) = PN+PS = P(Y(do(C=0.5)=0|C=1, Y=1)+P(Y(do(C=1)=1|Y=0,C=0.5)= 0.5 + 1 = 1.5

Case.2 PNS(1, 0) = PN+PS = P(Y(do(C=0)=0|C=1, Y=1)+P(Y(do(C=1)=1|Y=0,C=0) = 1 + 1 = 2

We note that Case.1 indicate the feature has more sufficiency than necessity.

We introduced a PNS risk to investigate the PNS value between the representation and Y. When the algorithm optimizes the objective function to minimize the PNS risk, it indicates that the representation we found contains more sufficient and necessary causal information.

Considering there are multiple choices of $c$ and $\bar{c}$ (like Example.2), we extend our algorithm by min-max optimization (Eq. 8 in our paper), which ait to find the worst $\bar{c}$ for $c$ with highest PNS risk.

**G.P. 2. Difference from other invariant learning methods:**

- i) Form the perspective of goals, while both our approach and invariant learning methods aim for domain generalization by discovering invariant features, our method further emphasises the importance of sufficient and necessary properties in the learned representations. The invariant feature doesn't mean sufficient and necessary. For example, the feature 'has cat legs' is invariant causal feature but it is unnecessary because having cat legs indicates it's a cat, but the absence of cat legs doesn't necessarily mean there is no cat in the image.


- ii) From the perspective of causal assumption, our assumption of the causal graph in Figure 1 is based on previous works on out-of-distribution (OOD) generalization, such as the causal graph in Liu et al. [1], and Ahuja et al. [2]. In Ahuja et al. [2], they conclude the assumptions in OOD generalization task as (1) fully informative invariant features (FIIF): Feature contains all the information about the label that is contained in input. (2) partially informative invariant features (PIIF). Our assumption $X \perp Y \mid C$ holds in the case of FIIF. As presented in [2] page 3, IRM [3] is based on the assumption of PIIF, where $X \perp Y \mid C$ does not hold and IRM fails in the case of FIIF.

  In our paper, we demonstrate that this paper only focuses on the causal assumption in Figure 1 where $X \perp Y \mid C$ (FIIF) holds.

  Other assumptions such as PIIF, confounders and anti-causal, will be solved by adding additional constraints. For example, in the assumption of PIIF, conditional independence should be represented as $Y \perp \mathcal{E} \mid C$ (PIIF). To satisfy Exogeneity, an objective function (e.g. invariant constraint in IRM) with additional domain label information is required.

- iii）From the perspective of objective function： The objective function of our method CaSN is presented in Eq. 8 in the paper. The objective function of IRM is

  $\min_{\phi} \sum_{e \in \mathcal{E}_{tr}} R^e(\phi)+\lambda | ∇ {w \mid w=1.0} R^e(w \cdot \phi)|^2,$ where the second term is invariant constraint.

  Our approach differs from the IRM-based methods on the final objective function for several reasons: 1) Because of the different assumptions, in our paper, there is no need to introduce additional domain label information. However, under the PIIF assumption where IRM operates [2], there's a need to introduce additional domain label information to compute the second term in the IRM objective function. Thus, IRM is only available for multi-domain scenarios, but our method can be used without providing domain information. 2) Our proposed method CaSN is built upon the PNS risk, while IRM is constructed on the ERM risk. From a causal perspective, PNS risk offers a more precise design compared to ERM, which allows CaSN to not only achieve invariant learning but also further discern sufficiency and necessary features.

We will add the discussion in the revision.

[1] Liu et. al., Learning causal semantic representation for out-of-distribution prediction

[2] Ahuja et al. Invariance principle meets information bottleneck for out-of-distribution generalization.

[3] Arjovsky et al. Invariant risk minimization.

[4] Joshi et al. Towards Mitigating Spurious Correlations in the Wild: A Benchmark & a more Realistic Dataset.

---

### Decision · Program_Chairs · 2023-09-21

**Decision:**

Accept (spotlight)

**Comment:**

The paper presents an interesting idea to capture causal features for a particular task: using the notion of necessary and sufficiency for an outcome variable. Reviewers appreciated the clean writing as well as the theoretical results.  The difference to the invariant property can be better explained, which I hope the authors will add to their camera-ready version. In addition, I recommend the authors to add the empirical experiments from their rebuttal to the main paper.